# p107 mediated mitochondrial function controls muscle stem cell proliferative fates

Debasmita Bhattacharya [1,2], Vicky Shah [1,3], Oreoluwa Oresajo [1,2] & Anthony Scimè [1,2,3 ✉]

Muscle diseases and aging are associated with impaired myogenic stem cell self-renewal and fewer proliferating progenitors (MPs). Importantly, distinct metabolic states induced by glycolysis or oxidative phosphorylation have been connected to MP proliferation and differentiation. However, how these energy-provisioning mechanisms cooperate remain obscure. Herein, we describe a mechanism by which mitochondrial-localized transcriptional co-repressor p107 regulates MP proliferation. We show p107 directly interacts with the mitochondrial DNA, repressing mitochondrial-encoded gene transcription. This reduces ATP production by limiting electron transport chain complex formation. ATP output, controlled by the mitochondrial function of p107, is directly associated with the cell cycle rate. Sirt1 activity, dependent on the cytoplasmic glycolysis product NAD$^+$, directly interacts with p107, impeding its mitochondrial localization. The metabolic control of MP proliferation, driven by p107 mitochondrial function, establishes a cell cycle paradigm that might extend to other dividing cell types.

[1] Stem Cell Research Group, York University, Toronto, ON M3J 1P3, Canada. [2] Molecular, Cellular and Integrative Physiology, Faculty of Health, York University, Toronto, ON M3J 1P3, Canada. [3] Department of Biology, York University, Toronto, ON M3J 1P3, Canada. ✉email: ascime@yorku.ca

In recent years, there has been an emerging realization that metabolism plays an active role in guiding and dictating how the cell will behave[1–3]. Central to this assertion is the interplay between metabolic states based on glucose metabolism by glycolysis in the cytoplasm with oxidative phosphorylation (Oxphos) in the mitochondria. Indeed, the cell cycle is dependent on the amount of total ATP generated from glycolysis and Oxphos[4,5].

Importantly, it is uncertain how glycolysis and Oxphos work together during proliferation to regulate the yield of ATP necessary for cell division. Proliferating cells customize methods to actively reduce Oxphos under conditions of higher glycolysis[6,7]. Fundamental to understanding this relationship is the cytoplasmic NAD$^+$/NADH ratio, which is a reflection of the energy generated by glycolysis. NADH is a by-product of glycolysis that can be indirectly transferred to the mitochondria via the Malate-Aspartate shuttle to be used in Oxphos as reducing equivalents. Alternatively, NADH might be oxidized to NAD$^+$ in the reaction that produces lactate from pyruvate, the end product of glycolysis.

NAD$^+$ is an important coenzyme in several metabolic pathways and has been implicated to be essential for proliferation[7–11]. It is an activator of seven sirtuin protein family members (Sirt1-7), which are deacetylases that target several proteins[12]. The sirtuins operate in specific cellular locations with nonredundant substrate preferences. Sirt3, Sirt4, and Sirt5 are located in the mitochondria, Sirt6 and Sirt7 in the nucleus, and Sirt1 and Sirt2 are found in both the nucleus and cytosol[13]. Of these, Sirt1 is the most well-studied family member operating as an energy sensor of the cytoplasmic NAD$^+$/NADH ratio[12]. It regulates metabolic homeostasis by enhancing mitochondrial metabolism through activation by cytoplasmic NAD$^+$ [11]. In contrast, decreased NAD$^+$/NADH and Sirt1 activity is associated with reduced mitochondrial function as observed in aging and metabolic diseases[11,14,15].

In the mitochondria, the capacity for ATP production is dependent on the five electron transport chain (ETC) complexes, which are made up of protein subunits derived from the nuclear and mitochondrial genomes[16]. The closed double-stranded circular mitochondrial DNA (mtDNA) encode genes located on both strands, which are referred to as the heavy (H) and light (L) strands. Their promoters are located in the large noncoding region termed the displacement loop (D-loop)[17]. The mitochondrial genome is essential to control energy generation by the ETC, as thirteen mitochondrial-encoded genes are functional components of four out of five ETC complexes. Crucially, mitochondrial genes are limiting for ATP production and inhibiting their transcription can block the cell cycle in G1[4,5,18,19].

It is uncertain if the mtDNA transcription initiation machinery comprises transcription factor B2 of mitochondria (Tfb2m) and mitochondrial DNA-directed RNA polymerase (Polrmt) or a three-component system that also includes transcription factor A of mitochondria (Tfam)[20–22]. Besides the initiation factors, the transcription machinery also includes the mitochondrial transcription elongation factor (Tefm), which ensures processivity and stabilization of the elongation complex and termination of transcription performed by mitochondrial transcription termination factor 1 (Mterf1)[17]. Multiple reports have suggested that transcription factors and co-transcriptional regulators that are typically functional in the nucleus also have regulatory functions in the mitochondria[23]. However, relative to nuclear gene expression, regulation of mitochondrial gene expression is poorly understood.

Mounting evidence has shown that Rb1 (Rb) and Rbl1 (p107), members of the retinoblastoma family of transcriptional corepressors, influence the energy status of cells and tissues[2,24–30]. p107 is ubiquitously expressed in proliferating cells and its overexpression has been shown to block cell cycle[31]. However, it is also implicated in controlling stem cell and progenitor metabolic fates[25,26,29]. Nonetheless, the fundamental mechanism that links p107 to metabolism and cell cycle has never been examined.

Key to understanding this relationship is the muscle resident myogenic stem cells, satellite cells (SCs), which are characterized by dynamic metabolic reprogramming during different stages of the differentiation process[32]. They use predominately Oxphos in quiescence, but their committed progeny, the myogenic progenitor (MP) cells up-regulate glycolysis to support proliferation[32,33]. In this study, MPs were used to uncover a control mechanism of energy generation that showcases the interplay between glycolysis and Oxphos in regulating cell proliferation. Intriguingly, we found that p107 accomplishes this by controlling ATP generation capacity in the mitochondria through a Sirt1 dependent mechanism. Thus, our findings highlight a crucial role for p107 linking the regulation of mitochondrial Oxphos to proliferation dynamics in MPs that is likely to extend to other cell types.

## Results

**p107 is localized in the mitochondria of myogenic progenitor (MP) cells.** By Western blotting cytoplasmic and nuclear cellular fractions, p107 was found in the cytoplasm of C2C12 myoblasts, an actively proliferating MP cell line we designated as "c2MPs" (Fig. 1a). Considering that emergent findings show that p107 can influence the metabolic state of progenitors[2], we assessed its potential metabolic role in the cytoplasm. As mitochondria are crucial in controlling metabolism, we assessed the presence of p107 within this organelle. Intriguingly, Western blot analysis demonstrated that p107 was in the mitochondrial fraction of the cytoplasm during proliferation, but almost absent at the outset of contact inhibition in confluent cultures we designated as "growth arrested" (Fig. 1b, Supplementary Fig. 1). Of the total p107 protein that was expressed in proliferating c2MPs, about 28% and 45% was present in the mitochondrial and cytoplasmic compartments, respectively (Supplementary Fig. 2). Unlike p107, in proliferating cells, family member Rb is not expressed in the mitochondria and Rbl2 (p130) is not expressed at all in any cellular compartments (Fig. 1c). Though p107 does not have an N-terminal mitochondrial targeting signal (mts), we found very strong scores (< 0.75) for putative internal mts-like sequences using the TargetP prediction algorithm[34], which normally predicts N-terminal presequences (Supplementary Fig. 3)[35,36].

To further assess for the presence of p107, we isolated various mitochondrial fractions utilizing hypotonic osmotic shock by varying sucrose concentrations[37]. Mitochondrial fractionation showed that p107 was not located at the outer membrane nor within the soluble inner membrane fraction whose soluble proteins were relinquished by decreasing the buffer molarity (Fig. 1d). However, it was localized within the matrix where the protein constituents are unavailable to trypsin for digestion compared to the soluble inner membrane proteins such as Cox4, as well as p107 from the whole cell lysate control (Fig. 1d).

The protein localization results were corroborated by confocal microscopy and subsequent analysis of generated z-stacks that showed p107 co-localized with the mitochondrial protein Cox4 and MitoTracker Red in c2MPs (Fig. 1e, Supplementary Fig. 4a). The specificity of immunocytochemistry was confirmed by immunofluorescence of p107 in p107 "knockout" c2MPs (p107KO c2MPs) generated by Crispr/Cas9 (Supplementary Fig. 5a). Moreover, p107 and Cox4 or MitoTracker Red fluorescence intensity peaks were matched on a line scanned RGB profile (Fig. 1f, Supplementary Fig. 4b) and orthogonal projection showed co-localization in the XY, XZ, and YZ planes (Fig. 1g, Supplementary Fig. 4c). The same assays were used to

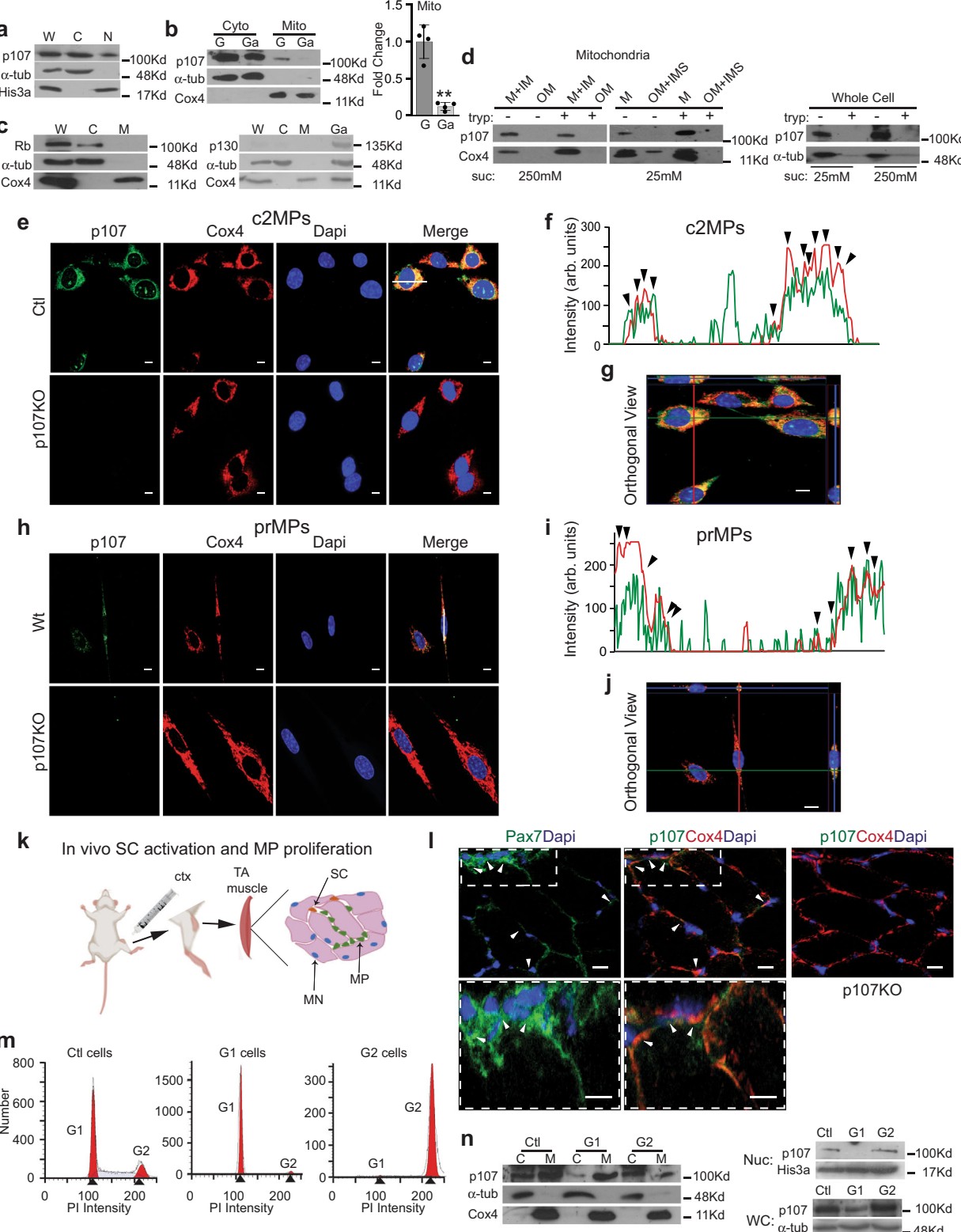

confirm these findings in proliferating primary wild type (Wt) MPs (prMPs) and p107 genetically deleted prMPs (p107KO prMPs) isolated from Wt and p107KO mouse skeletal muscles, respectively (Fig. 1h, i, j, Supplementary Fig. 4d–f). In vivo, in a model of regenerating skeletal muscle caused by cardiotoxin injury, we also found that p107 co-localized with Cox4 in proliferating MPs, which express the myogenic stem cell marker Pax7 (Fig. 1k, l, Supplementary Fig. 6). Together, these data

showcase that p107 localizes in the mitochondria of MPs from primary and muscle cell lines, suggesting that it might have an important mitochondrial function in the actively dividing cells.

We determined during cell proliferation that p107 was localized in the mitochondria by Western blotting cellular fractions of cells that were almost entirely in the G1 or G2 phases of the cell cycle (Fig. 1m, Supplementary Fig. 7, 8). We found that in G1 p107 was expressed in the mitochondria and

**Fig. 1 p107 is localized in the mitochondria of proliferating myogenic progenitors. a** Representative Western blot of whole cell (W), cytoplasmic (C), and nuclear (N) fractions for p107, cytoplasm loading control α-tubulin (α-tub) and nucleus loading control, His3a, during proliferation. **b** Representative Western blot and graphical representation of cytoplasmic (Cyto) and mitochondrial (Mito) c2MP fractions for p107, α-tub and mitochondria loading control, Cox4, during proliferation (G) and growth arrest (Ga). Graphical data are presented as mean values ± SD ($n = 4$ biologically independent samples). Two-tailed unpaired Student's $t$ test **$p = 0.00343$. **c** Representative Western blot of proliferating whole cell (W) cytoplasmic (C), mitochondrial (M) and growth arrested (Ga) cells for Rb, p130, α-tub and Cox4. **d** Representative Western blot of c2MP whole cell and mitochondrial fractions including outer membrane (OM), inner membrane (IM), soluble inner membrane (IMS) and matrix (M) that were isolated in 250 mM or 25 mM sucrose buffer, treated and untreated with trypsin. Confocal immunofluorescence microscopy for p107, Cox4, Dapi and Merge of proliferating **e** Control (Ctl) and genetically deleted p107 (p107KO) c2MPs and **h** wild type (Wt) and p107KO primary (pr) MPs (scale bar 10 μm). **f, i** A line was drawn through a representative cell to indicate relative intensity of RGB signals with the arrowheads pointing to areas of concurrent intensities **g, j** an orthogonal projection was generated by a Z-stack (100 nm interval) image set using the ZEN program (Zeiss) in the *XY*, *XZ*, and *YZ* planes (scale bar 10 μm). **k** Schematic for inducing in vivo satellite cell (SC) activation and MP proliferation in the tibialis anterior (TA) muscle with cardiotoxin (ctx) injury (MN is myonuclei). **l** Confocal immunofluorescence microscopy merged image of wild type (Wt) TA muscle section 2 days post ctx injury for Pax7 and Dapi and p107, Cox4 and Dapi and for p107KO TA muscle section for p107, Cox4 and Dapi (scale bar 20μm). Arrows denote Pax7 [+]p107[+]Cox4[+] cells. **m** Cell cycle histograms using flow cytometry for cell number versus propidium iodide (PI) intensity for c2MPs that are proliferating (Ctl), predominantly in G1 and G2 phase of the cell cycle. **n** Representative Western blot of cytoplasmic (C), mitochondrial (M), nuclear (Nuc) and whole-cell lysates (WC) for p107, α-tub, Cox4 and His3a from Ctl, G1 and G2 cells. Source data are provided as a Source Data file.

absent in the nucleus, contrary to G2 where it is expressed both in mitochondria and the nucleus (Fig. 1n). This data suggests that p107 might have an exclusive mitochondrial and not a nuclear function during G1 phase of the cell cycle.

**p107 interacts at the mtDNA.** To find the functional consequence of p107 mitochondrial matrix localization we assessed if it interacts at the mtDNA to repress mitochondrial gene expression similar to its role as a co-transcriptional repressor on nuclear promoters[31]. We evaluated this potential by performing quantitative chromatin immunoprecipitation (qChIP) analysis using primer sets that span the D-loop regulatory region of isolated mitochondria from c2MPs. qChIP revealed that p107 interacted at the mtDNA of growing c2MPs (Fig. 2a). mtDNA interaction was negligible in growth-arrested cells when p107 levels are deficient in the mitochondria and not detected in p107KO c2MPs (Fig. 2a).

As p107 does not directly interact with DNA we assessed potential interacting transcription factors in the mitochondria. We evaluated the potential mitochondrial role for the putative mitochondrial transcription factor Tfam, as well as the p107 interacting nuclear transcription factors E2f4 and E2f5[31]. By Western blotting, we found that E2f4 was present in proliferating mitochondrial lysates, in contrast to negligible levels of Tfam and the complete absence of E2f5 (Fig. 2b). Moreover, by ChIP of proliferating c2MPs, we found that E2f4 interacted at the mtDNA and not Tfam; however, the opposite pattern of interaction was found during growth arrest (Fig. 2c). Mitochondrial initiation factors Polrmt and Tfb2m were not ChIPed because their mouse-specific antibodies capable of immunoprecipitation are not commercially available nor has their use been published. Intriguingly, these results suggest that E2f4 is a possible binding partner for p107 at the mtDNA during the proliferation of c2MPs.

**p107 is associated with lowered mtDNA encoded gene expression.** We next appraised if p107 interaction at the mtDNA might influence mitochondrial-encoded gene expression. For this we assessed 4 of 13 mitochondrial genes that are subunits of 3 out of 5 ETC complexes. qPCR analysis showed that during proliferation when p107 is abundant in the mitochondria, there was significantly less expression of the mitochondrial-encoded genes compared to growth-arrested cells (Fig. 2d). The importance of p107 to mitochondrial gene expression was confirmed with p107KO c2MPs and prMPs, which both exhibited

significantly increased mitochondrial-encoded gene expression in the genetically deleted cells compared to their controls (Fig. 2e).

We eliminated the potential cofounding effect of the cell cycle state in comparing control and p107 knockout cells. This was undertaken by harmonizing their cell cycles so that the number of cells in any cell cycle phase was identical for both cell types. For this, cells were synchronized to the G1 phase of the cell cycle (Supplementary Fig. 9), a time point when p107 is present in the mitochondria (Fig. 1n). As with the asynchronous cells (Fig. 2e), we found that the p107KO cells had significantly greater mitochondrial gene expression compared to controls with identical cell cycle profiles (Fig. 2f).

The increased mitochondrial gene expression in growth-arrested and p107KO cells corresponded to augmented ETC complex formation revealed by Western blot analysis of the relative protein levels of the 5 Oxphos complexes (Fig. 2g). The increased ETC complex formation may not only signify differences in mtDNA transcription, as we also found that relative mtDNA copy number was significantly greater in p107KO cells compared to controls (Fig. 2h).

**p107 mitochondrial function is associated with Oxphos capacity.** As mitochondrial gene expression is limiting for ETC complex formation, we used two methods to gauge if p107 interaction at the mtDNA was associated with the capacity of mitochondria to generate ATP. First, using a luminescence assay, ATP generation potential capacity and rate were determined from isolated mitochondria. To exclude the potential contribution of mitochondria biogenesis, measurements were normalized to mitochondrial protein content. We found that the potential rate and capacity of ATP formation from isolated mitochondria of growth-arrested cells and p107KO c2MPs were significantly higher than proliferating and control c2MPs, respectively (Fig. 2i, j and Supplementary Table 1a, b). This corresponded to the increase in mitochondrial gene expression profile and ETC complex formation (Fig. 2e–g). Though, mtDNA copy number, which typically is associated with mitochondria content, was significantly greater in p107KO cells (Fig. 2h), the differences in ATP generation rate and capacity between control and p107KO cells are reflective of differences in ETC levels per mitochondria rather than mitochondria number.

Second, we confirmed these results with live cell metabolic analysis using Seahorse, which showed that both growth-arrested cells and p107KO c2MPs had significantly enhanced production of mitochondrial ATP generated compared to the proliferating

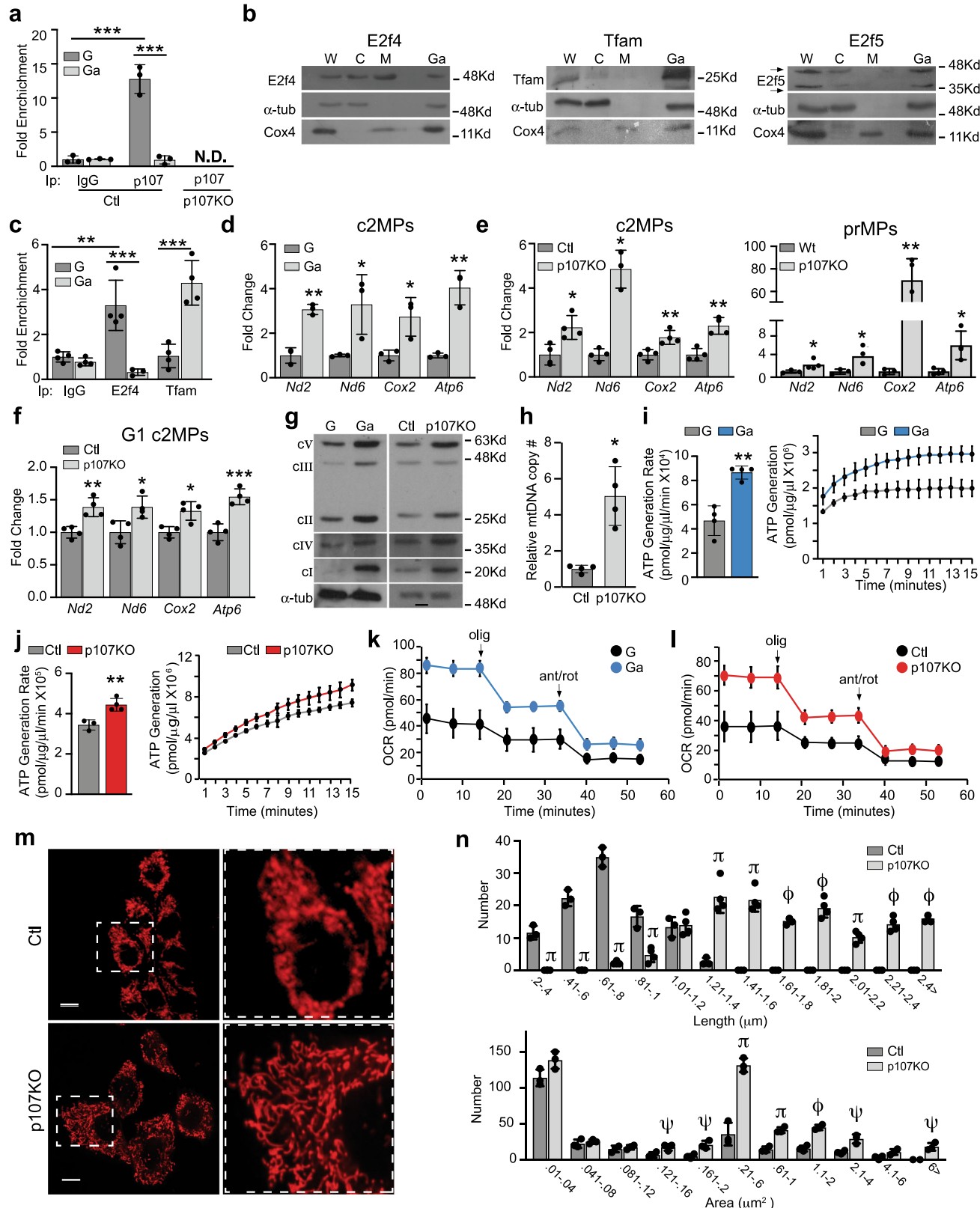

and control c2MPs, respectively (Fig. 2k, l, Supplementary Fig. 10). As Oxphos is known to increase mitochondrial fusion causing elongation, we assessed a potential effect on mitochondrial morphology[38,39]. Interestingly, we found that p107KO c2MPs had an elongated mitochondrial network (Fig. 2m) composed of mitochondria with significantly increased length

and area (Fig. 2n, Supplementary Table 2). Aside from mitochondrial fusion the elongation could also be due to increased numbers of mitochondria, which under the microscope may appear as elongated structures. Together, these data suggest that p107 has repressor activity when interacting at the mtDNA regulating mtDNA copy number and/or gene expression to

**Fig. 2 p107 down regulates mitochondrial ATP generation capacity. a** Graphical representation of relative p107 and IgG mitochondrial DNA occupancy by qChIP analysis during proliferation (G) and growth arrest (Ga) in control (Ctl) and p107KO c2MPs. Data are presented as mean values ± SD ($n = 3$ biologically independent samples). Two-way Anova with post hoc Tukey ***$p < 0.001$ G (IgG vs. p107) and *** $p < 0.001$ ChIP p107 (G vs Ga), N.D. (not detected). **b** Representative Western blot of proliferating whole cell (W), cytoplasmic (C), mitochondrial (M) and growth arrested (Ga) cells for E2f4, Tfam, E2f5, α-tub and Cox4 lysates. **c** Graphical representation of relative E2f4, Tfam and IgG mitochondrial DNA occupancy by qChIP analysis during proliferation (G) and growth arrest (Ga) in control c2MPs, Data are presented as mean values ± SD ($n = 4$ biologically independent samples). Two-way Anova with post hoc Tukey, **$p = 0.0014$ G (IgG vs. E2f4), ***$p < 0.001$ ChIP E2f4 (G vs Ga) and ***$p < 0.001$ ChIP Tfam (G vs Ga). Gene expression analysis by RT-qPCR of mitochondrial-encoded genes *Nd2*, *Nd6*, *Cox2*, and *Atp6* for: **d** c2MPs during G and Ga ($n = 3$ biologically independent samples), **e** Ctl and p107KO c2MPs and Wt and p107KO prMPs during proliferation ($n = 4$ biologically independent samples) and **f** Ctl and p107KO c2MPs in G1 phase of the cell cycle ($n = 4$ biologically independent samples). Data are presented as mean values ± SD. Two-tailed unpaired Student's *t* test: G and Ga *$p = 0.0415$ (*Nd6*), 0.0291 (*Cox2*), **$p = 0.0012$ (*Nd2*), 0.0025 (*Atp6*); Ctl and p107KO: *$p = 0.0139$ (*Nd2*), 0.0114 (*Nd6*), **$p = 0.00186$ (*Atp6*), 0.00834 (*Cox2*) and Wt and p107KO *$p = 0.0418$ (*Nd2*), 0.0371 (*Nd6*), 0.0330 (*Atp6*), **$p = 0.0058$ (*Cox2*), and Ctl and p107KO c2MPs in G1 phase *$p = 0.0195$ (*Nd6*), 0.0110 (*Cox2*), **$p = 0.0062$ (*Nd2*), ***$p < 0.001$ (*Atp6*). **g** Representative Western blot of G compared to Ga, and Ctl compared to p107KO c2MPs for each electron transport chain complex (**c**) subunits, cI (NDUFB8), cII (SDHB), cIII (UQCRC2), cIV (MTCO) and cV (ATP5A). **h** Relative mitochondrial DNA (mtDNA) copy number (#) for control (Ctl) and p107KO c2MPs. Data are presented as mean values ± SD ($n = 4$ biologically independent samples). Two-tailed unpaired Student's *t* test, *$p = 0.0145$. ATP generation rate and capacity over time for isolated mitochondria from (**i**) G and Ga c2MPs and (**j**) Ctl and p107KO c2MPs. Data are presented as mean values ± SD ($n = 4$ biologically independent samples). Two-tailed unpaired Student's *t* test, G and Ga c2MPs **$p = 0.0037$ and Ctl and p107KO c2MPs **$p = 0.0057$. For capacity significance statistics see Supplementary Table 1a, b. Live cell metabolic analysis by Seahorse for oxygen consumption rate (OCR) with addition of oligomycin (olig) and antimycin A (ant), rotenone (rot) for (**k**) G and Ga c2MPs and (**l**) Ctl and p107KO c2MPs, Data are presented as mean values ± SD ($n = 8$ G, 10 Ga, 4 Ctl and 6 p107KO biologically independent samples). **m** Representative confocal microscopy live cell image of stained mitochondria (MitoView Red) for Ctl and p107KO c2MPs (scale bar 10 μm). **n** Graphical representation of mitochondria number per grouped lengths and areas using image J quantification. Data are presented as mean values ± SD ($n = 3$ biologically independent samples). Two-tailed unpaired Student's *t* test, significance denoted by ψ = $p < 0.05$, π = $p < 0.01$, φ = ***$p < 0.001$. Detailed statistics in Supplementary Table 2. Source data are provided as a Source Data file.

reduce the capacity to produce ETC complex subunits, which influences the mitochondria potential for ATP generation.

**Glycolytic flux controls p107 mitochondrial function**. The reliance of glycolysis versus Oxphos during the cell cycle can be driven by nutrient accessibility. Thus, we evaluated how nutrient availability influences p107 compartmentalization during proliferation. We assessed the effect of glucose metabolism on p107 mitochondrial localization by growing c2MPs in stripped media with glucose as the sole nutrient. Western blot analysis of isolated mitochondria revealed that p107 translocation into mitochondria was glucose concentration dependent (Fig. 3a). Its presence was negligible in mitochondria when cells were grown in 5.5 mM glucose, but there was a significant increase of p107 localization in mitochondria at a higher glucose concentration (25 mM). The presence of p107 in the mitochondria was inversely associated with the amount in the nucleus (Fig. 3a) and was not affected by an increase in p107 gene expression (Supplementary Fig. 11).

The increased level of mitochondrial p107 in c2MPs grown in higher glucose concentration corresponded to significantly increased mtDNA binding (Fig. 3b). This was associated with a significant decrease in mitochondrial-encoded gene expression in both c2MPs and prMPs grown in high glucose concentration (Fig. 3c, Supplementary Fig. 12). However, increasing glucose concentration in the stripped media had no effect on mitochondrial gene expression in the p107KO c2MPs and p107KO prMPs compared to their controls (Fig. 3c, Supplementary Fig. 12).

Importantly, our results also verified that changes in mitochondrial gene expression in MPs grown in varying glucose concentrations are not influenced by differences in mitochondrial biogenesis. By using qPCR, the gene expression of key markers of mitochondrial biogenesis *Pgc-1α* and *Mfn2* were not detected and there were no differences for expression of *Ant1* and *Nrf2* in c2MPs grown in 5.5 mM compared to 25 mM glucose (Fig. 3d). Moreover, the relative mtDNA copy number remained unchanged when grown in the different glucose concentrations, in contrast to MPs treated with the mitochondrial biogenesis activator AICAR (Fig. 3e). Additionally, we found that in p107KO cells the relative levels of mtDNA did not change when grown in the different glucose concentrations (Fig. 3f). Together, these data show that within the time period of the experiments, the regulation of mitochondrial-encoded gene expression by glucose concentrations is not due to altered mtDNA number and/or mitochondrial biogenesis.

We investigated if the reduction of mitochondrial gene expression with increased glucose availability influenced the potential mitochondrial ATP synthesis. In isolated mitochondria, we determined that there was a significant decrease in the potential rate and capacity of ATP generation when cells were grown in 25 mM compared to 5.5 mM glucose (Fig. 3g, Supplementary Table 1c). This corresponds to when p107 is in the mitochondria compared to low glucose where it is barely present. However, in the p107KO c2MPs, where mitochondrial gene expression was not influenced by glucose (Fig. 3c), the potential mitochondrial ATP rate and generation capacity were also unaffected by glucose (Fig. 3h). These data suggest that p107 is controlled by glucose metabolism to influence the potential mitochondrial energy production through regulation of the mitochondrial gene expression. Together, the results showcase how glucose concentration alone, without the confounding contribution of other key nutrients, affects ATP generation potential.

**NAD⁺/NADH regulates p107 mitochondrial function**. As glucose metabolism affects the $NAD^+/NADH$ redox balance, we determined if this might impact p107 mitochondrial localization and function. We found that c2MPs grown in 5.5 mM compared to 25 mM glucose in stripped media had significantly increased cytoplasmic $NAD^+/NADH$ ratio (Fig. 3i). This suggests that the cytoplasmic $NAD^+/NADH$ ratio might influence mitochondrial gene expression and ATP generation capacity via a p107 dependent mechanism.

We manipulated the cytoplasmic $NAD^+/NADH$ energy flux in a glucose concentration independent manner to find the importance of the redox potential to p107 function. We added oxamate (ox) or dichloroacetic acid (DCA) or used RNAi to knockdown Ldha (Supplementary Fig. 13a) to decrease the $NAD^+/NADH$ ratio (Fig. 3j, Supplementary Fig. 13b, 14a).

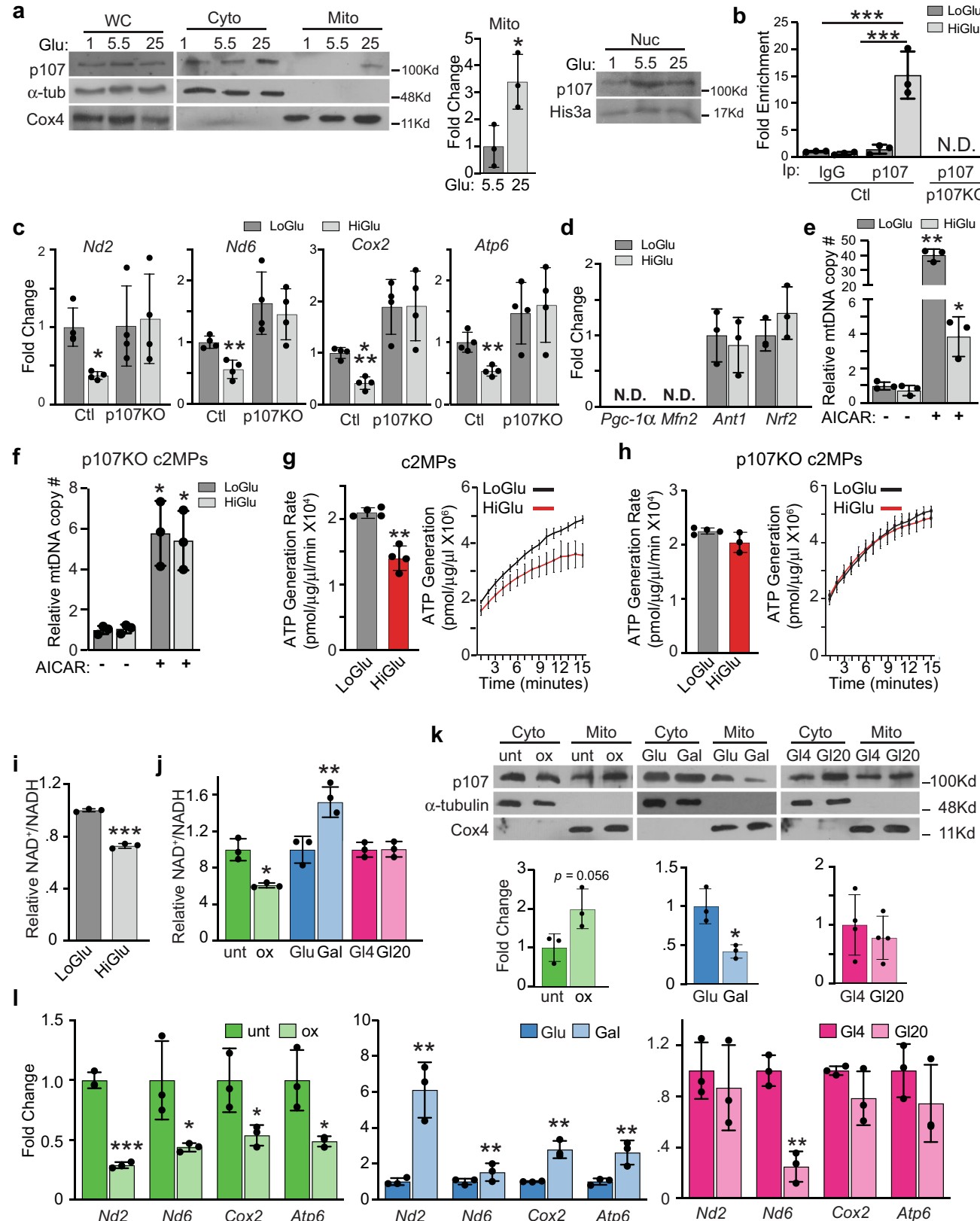

Conversely, we grew cells in the presence of galactose instead of glucose, which increases the NAD$^+$/NADH ratio and cultured cells in varying amounts of glutamine in stripped media, which had no effect on the cytoplasmic NAD$^+$/NADH ratio (Fig. 3j). As anticipated, Western blot analysis of c2MPs treated with ox, DCA or Ldha RNAi showed significantly increased p107 levels in the

mitochondria (Fig. 3k, Supplementary Figs. 13c, 14b). On the other hand, galactose-treated cells had significantly less p107 in the mitochondria compared to untreated controls and cells treated with glutamine showed no difference in the level of p107 mitochondrial localization (Fig. 3k). These results suggest that p107 mitochondrial localization is based on the cytoplasmic

**Fig. 3 p107 discerns the NAD$^+$/NADH ratio to regulate mitochondrial gene expression and Oxphos. a** Representative Western blot and graphical representation of whole cell (WC), cytoplasmic (Cyto), mitochondrial (Mito) and nuclear (Nuc) fractions from c2MPs grown in stripped media (SM) containing only 1.0 mM, 5.5 mM or 25 mM glucose (Glu) for p107, α-tub, Cox4, and His3a. Graphical data are presented as mean values ± SD ($n = 3$ biologically independent samples). Two-tailed unpaired Student's $t$ test, *$p = 0.0314$. **b** Graphical representation of relative p107 and IgG mitochondrial DNA occupancy by qChIP analysis for control (Ctl) and p107KO c2MPs grown in SM containing only 5.5 mM (LoGlu) or 25 mM (HiGlu) glucose. Data are presented as mean values ± SD ($n = 3$ biologically independent samples). Two-way Anova with post hoc Tukey, ***$p < 0.001$ (LoGlu IgG vs. HiGlu p107) and ***$p < 0.001$ ChIP p107 (LoGlu vs HiGlu), N.D. is not detected. **c** RT-qPCR of *Nd2*, *Nd6*, *Cox2*, and *Atp6* for Ctl and p107KO c2MPs grown in SM containing LoGlu or HiGlu. Data are presented as mean values ± SD ($n = 4$ biologically independent samples). Two-tailed unpaired Student's $t$ test, HiGlu vs. LoGlu Ctl, *$p = 0.0132$ (*Nd2*), **$p = 0.0041$ (*Nd6*), 0.0041 (*Atp6*), ***$p < 0.001$ (*Cox2*). **d** RT-qPCR of *Pgc-1α*, *Mfn2*, *Ant1*, and *Nrf2* for c2MPs grown in SM containing LoGlu or HiGlu. Data are presented as mean values ± SD ($n = 3$ biologically independent samples) Two-tailed unpaired Student's $t$ test, N.D. (not detected). Relative mitochondrial DNA (mtDNA) copy number (#) for (**e**) c2MPs and (**f**) p107KO c2MPs grown in SM containing LoGlu or HiGlu in the absence or presence of 5-aminoimidazole-4-carboxamide ribonucleotide (AICAR), Data are presented as mean values ± SD ($n = 3$ biologically independent samples). Two-tailed unpaired Student's $t$ test, for c2MPs *$p = 0.0454$ (HiGlu vs. HiGlu AICAR), **$p = 0.0035$ (LoGlu vs. LoGlu AICAR) and for p107KO c2MPs *$p = 0.0336$ (LoGlu vs. LoGlu AICAR), 0.0333 (HiGlu vs. HiGlu AICAR). ATP generation rate and capacity over time for isolated mitochondria from (**g**) c2MPs and (**h**) p107KO c2MPs, grown in SM containing LoGlu or HiGlu. Data are presented as mean values ± SD ($n = 4$ biologically independent samples). Two-tailed unpaired Student's $t$ test, for c2MP rate **$p = 0.0025$. For capacity significance statistics see Supplementary Table 1c. **i** NAD$^+$/NADH ratio for c2MPs grown in SM containing LoGlu or HiGlu. Data are presented as mean values ± SD ($n = 3$ biologically independent samples). Two-tailed unpaired Student's $t$ test, ***$p < 0.001$. **j** NAD$^+$/NADH ratio for c2MPs untreated (unt) or treated with 2.5 mM oxamate (ox), grown in 25 mM glucose (Glu) or 10 mM galactose (Gal) or with varying glutamine concentrations of 4 mM (Gl4) or 20 mM (Gl20). Data are presented as mean values ± SD ($n = 3$ biologically independent samples). Two-tailed unpaired Student's $t$ test, *$p = 0.0251$, **$p = 0.0079$. **k** Representative Western blots and graphical representation of mitochondrial (Mito) fractions of cells treated as in (**j**). Data are presented as mean values ± SD ($n = 3$ biologically independent samples). Two-tailed unpaired Student's $t$ test, $p = 0.0560$ ox and *$p = 0.0143$ Gal. **l** RT-qPCR analysis of cells treated as in (**j**). Data are presented as mean values ± SD ($n = 3$ biologically independent samples). Two-tailed unpaired Student's $t$ test; for ox *$p = 0.0394$ (*Nd6*), 0.0466 (*Cox2*), 0.0263 (*Atp6*), ***$p < 0.001$ (*Nd2*) for Gal **$p = 0.0079$ (*Nd2*), 0.0075 (*Nd6*), 0.0049 (*Cox2*), 0.0065 (*Atp6*) and for Gl20 **$p = 0.0016$ (*Nd6*). Source data are provided as a Source Data file.

NAD$^+$/NADH ratio. Mitochondrial gene expression levels with the different treatments corresponded with p107 mitochondrial localization established by the cytoplasmic NAD$^+$/NADH ratio (Fig. 3l, Supplementary Fig. 13d, 14c). Indeed, significantly higher mitochondrial gene expression levels were observed with higher cytoplasmic NAD$^+$/NADH when cells were grown with galactose compared to glucose, whereas significantly lower mitochondrial gene expression was evident when cells were treated with ox or DCA or had an Ldha knockdown that caused decreased cytoplasmic NAD$^+$/NADH (Fig. 3l, Supplementary Figs. 13d, 14c). No significant gene expression changes were present when cells were grown solely with varying amounts of glutamine, except *Nd6*, which is the only mitochondrial gene expressed from the L-strand of mtDNA (Fig. 3l). Together, these results show that p107 acts indirectly as an energy sensor of the cytoplasmic NAD$^+$/NADH ratio that might influence the potential ATP produced from the mitochondria, as a consequence of regulating mitochondrial gene expression.

**Sirt1 directly regulates p107 mitochondrial function.** As the NAD$^+$ dependent Sirt1 deacetylase is an energy sensor of the NAD$^+$/NADH ratio found in the cytoplasm and targets several transcription factors and co-activators[12], we evaluated if it potentially regulates p107. Importantly, reciprocal immunoprecipitation/Western blot analysis on c2MP whole cell as well as cytoplasmic lysates for endogenous p107 and Sirt1 showed that they directly interacted (Fig. 4a, Supplementary Fig. 15). No interactions were apparent in p107KO and Sirt1 genetically deleted (Sirt1KO) c2MPs obtained by Crispr/Cas9 (Fig. 4a, Supplementary, 5b, 15).

We next assessed if Sirt1 activity affected p107 mitochondrial function. Unlike c2MPs grown in 5.5 mM glucose that exhibit relocation of p107 from the mitochondria (Fig. 3a), Sirt1KO cells did not alter p107 mitochondrial localization (Fig. 4b) nor influenced its gene expression, including when grown in different glucose concentrations (Supplementary Fig. 16a, b). p107 relocation was associated with significantly decreased mtDNA encoded gene expression for Sirt1KO cells grown in 5.5 mM

glucose compared to controls and no differences between Sirt1KO cells grown in 5.5 mM or 25 mM glucose (Fig. 4c). The decreased mitochondrial gene expression in Sirt1KO cells grown in 5.5 mM glucose corresponded with a reduced mitochondrial ATP generation rate and capacity compared to control cells (Fig. 4d, Supplementary Table 1d). There were no differences in ATP generation rate or capacity between Sirt1KO cells grown in 5.5 mM or 25 mM glucose (Supplementary Fig. 17).

We determined if Sirt1 activity is necessary for p107 mitochondrial function. Inhibition of Sirt1 activity by nicotinamide (nam) increased p107 mitochondrial localization (Fig. 4e) and not its gene expression levels (Supplementary Fig. 16c) that was concomitant with increased mtDNA interaction (Fig. 4f). This was associated with decreased mitochondrial gene expression, whereas nam treatment of p107KO and Sirt1KO c2MPs had no effect on the mitochondrial gene expression (Fig. 4g). This suggests that Sirt1 control of mitochondrial gene expression is dependent on p107. As expected, both the potential ATP generation rate and capacity of isolated mitochondria were reduced with Sirt1 attenuated activity (Fig. 4h, Supplementary Table 1e). On the contrary, Sirt1KO c2MPs showed no difference in ATP generation capacity and rate, and no changes in mitochondrial gene expression profile (Fig. 4g, i). Moreover, the reduction of Oxphos capacity by attenuated Sirt1 activity was not due to an apparent decrease in mitochondrial biogenesis, as nam treatment did not influence the mtDNA copy number (Fig. 4j).

Next, we activated Sirt1 for a short period of time to minimize any potential activity on pro-oxidative mitochondrial output caused by additional factors[12]. For this, c2MPs treated with Sirt1 activator, srt1760, for 3 h resulted in decreased p107 protein levels within the mitochondria (Fig. 4k). The activation of Sirt1 also reduced p107 interaction at the mtDNA (Fig. 4l) and enhanced the mitochondrial gene expression, but not in p107KO and Sirt1KO cells (Fig. 4m). The increase in gene expression with Sirt1 activation within this short time window was associated with increased ATP generation (Fig. 4n). Importantly, relative mtDNA copy number was not affected by this short-term treatment (Fig. 4o), suggesting that mitochondrial gene

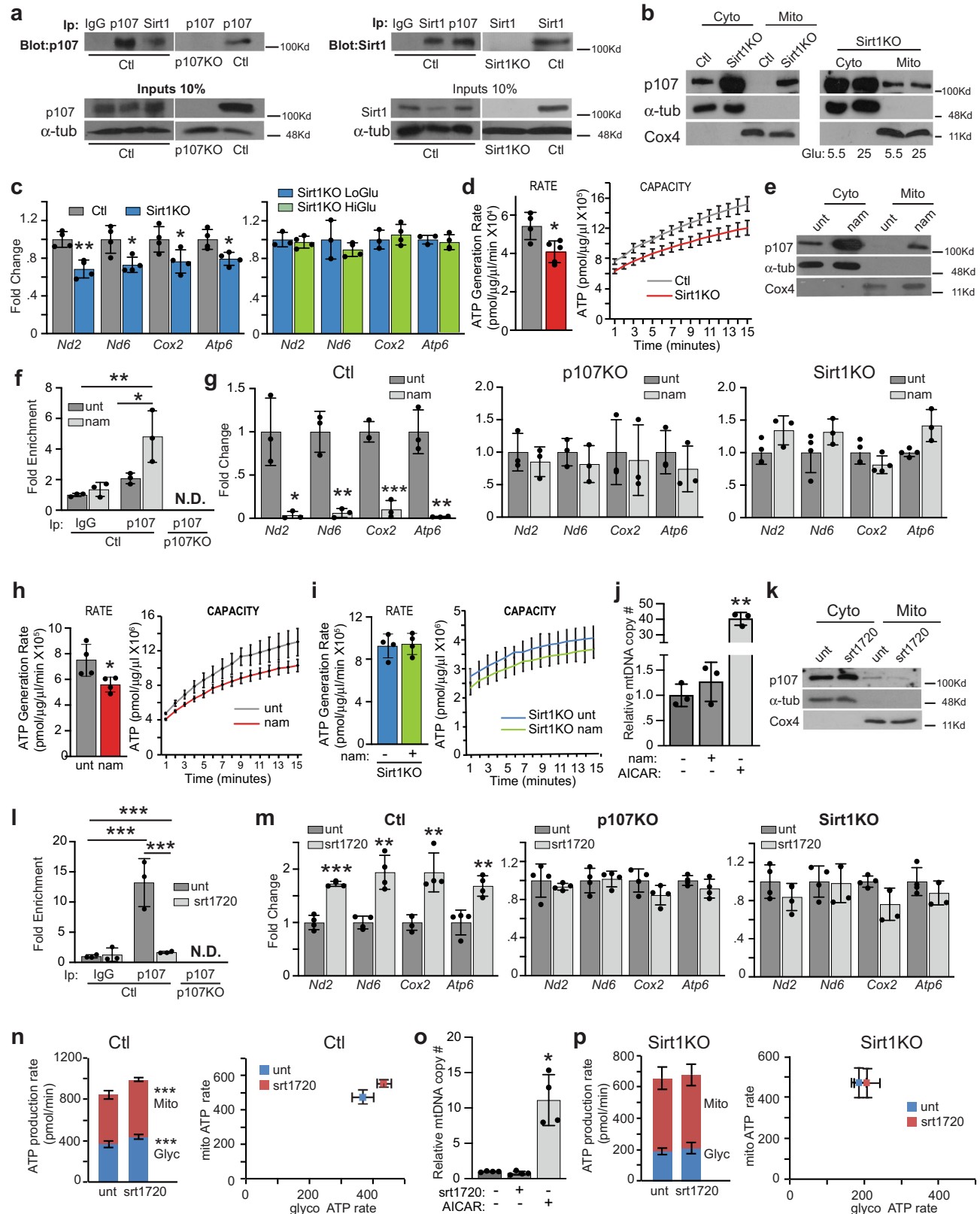

expression and not mtDNA copy number influenced differences in ATP generation. No differences in ATP generation rate and capacity were observed in Sirt1KO c2MPs (Fig. 4p). Moreover, we confirmed the importance of Sirt1 activity to p107 mitochondrial function by using resveratrol at low (Supplementary Fig. 18, Supplementary Table 1f) and high doses (Supplementary Fig. 19,

Supplementary Table 1g) that indirectly activate and inhibit Sirt1 activity, respectively.

We further assessed the role of Sirt1 activity on p107 localization by overexpressing c2MPs with Ha tagged p107 in the presence of full-length Sirt1 (Sirt1fl) or dominant-negative Sirt1 (Sirt1dn). Similar to Sirt1KO cells, Sirt1dn had no effect on

**Fig. 4 p107 is regulated by Sirt1. a** Immunoprecipitation/Western blots for p107 and Sirt1 in control (Ctl) and p107KO c2MPs. **b** Representative Western blot of cytoplasmic (Cyto) and mitochondrial (Mito) fractions of Ctl and Sirt1KO c2MPs grown in stripped media with 5.5 mM and 25 mM glucose. **c** RT-qPCR of *Nd2, Nd6, Cox2* and *Atp6* for Ctl and Sirt1KO c2MPs and Sirt1KO c2MPs grown in stripped media with 5.5 mM (LoGlu) and 25 mM (HiGlu) glucose. Data are presented as mean values ± SD ($n = 4$ biologically independent samples). Two-tailed unpaired Student's *t* test, *$p = 0.0261$ (*Nd6*), 0.0431 (*Cox2*), 0.0211 (*Atp6*), **$p = 0.0026$ (*Nd2*). **d** Isolated mitochondrial ATP generation rate and capacity over time for Ctl and Sirt1KO c2MPs grown in 5.5 mM glucose. Data are presented as mean values ± SD ($n = 4$ biologically independent samples). Two-tailed unpaired Student's *t* test, *$p = 0.0228$. For capacity significance statistics see Suppl. Table 1d. **e** Representative Western blot of Cyto and Mito fractions of c2MPs untreated (Ctl) or treated with Sirt1 inhibitor nicotinamide (nam) or (**k**) activator srt1720. Graphical representation of relative p107 and IgG mitochondrial DNA occupancy by qChIP analysis in Ctl and p107KO c2MPs untreated (unt) or treated with (**f**) nam and (**l**) srt1720. Data are presented as mean values ± SD ($n = 3$ nam and ctl and 4 srt1720 and ctl biologically independent samples). Two-way Anova with post hoc Tukey *$p = 0.0232$ ChIP p107 (unt vs. nam),**$p = 0.0035$ (IgG unt vs. p107 nam), ***$p < 0.001$ (IgG unt vs. p107 unt), (IgG unt vs. p107 srt1720) and ChIP p107 (unt vs. srt1720). RT-qPCR of *Nd2, Nd6, Cox2,* and *Atp6* for Ctl, p107KO and Sirt1KO c2MPs unt and treated with (**g**) nam and (**m**) srt1720. Data are presented as mean values ± SD (n = 3 for nam and ctl and 4 for srt1720 and ctl biologically independent samples). Two-tailed unpaired Student's *t* test. For nam treated cells *$p = 0.0129$ (*Nd2*), **$p = 0.0026$ (*Nd6*), 0.0026 (*Atp6*), ***$p < 0.001$ (*Cox2*) srt1720 treated cells, **$p = 0.0059$ (*Nd6*), 0.0091 (*Cox2*), 0.0042 (*Atp6*), ***$p < 0.001$ (*Nd2*). Isolated mitochondria ATP generation rate and capacity over time for **h** Ctl and **i** Sirt1KO c2MPs treated and untreated with nam. Data are presented as mean values ± SD ($n = 4$ biologically independent samples). Two-tailed unpaired Student's *t* test, *$p = 0.0480$. For ATP capacity statistics see Suppl. Table 1e. **j** Mitochondria DNA (mtDNA) copy number (#) for c2MPs with and without AICAR and nam. Data are presented as mean values ± SD (n = 3 biologically independent samples). One-way Anova with Tukey post hoc test, **$p = 0.0035$. ATP production from mitochondria (Mito) and glycolysis (Gly) for (**n**) Ctl ($n = 8$ biologically independent samples) and (**p**) Sirt1KO c2MPs ($n = 9$ biologically independent samples) treated and untreated with srt1720. Data are presented as mean values ± SD. Two-tailed unpaired Student's *t* test, ***$p < 0.001$. **o** mtDNA copy # for c2MPs with and without AICAR and srt1720. Data are presented as mean values ± SD ($n = 4$ biologically independent samples). Two-tailed unpaired Student's *t* test, *$p = 0.0111$. Source data are provided as a Source Data file.

preventing p107 mitochondria localization when grown in 5.5 mM glucose as opposed to Sirt1fl (Compare Fig. 4b and Supplementary Fig. 20). As expected, the Sirt1fl overexpression did not alter p107 localization in cells grown in 25 mM glucose (Compare Fig. 4b and Supplementary Fig. 21). Also, there was significantly less mitochondrial-encoded gene expression in Sirt1dn cells grown in 5.5 mM glucose compared to Sirt1fl (Compare Fig. 4c, Supplementary Fig. 22). Together, these results suggest that p107 is influenced directly by Sirt1 activity to localize within the cytoplasm, de-repressing mitochondrial gene expression and hence increasing ATP generation capacity.

**p107 directs cell cycle rate through management of Oxphos generation.** We considered how this metabolic role for p107 might influence MP behavior in vivo by assessing skeletal muscle regeneration caused by cardiotoxin injury (Fig. 5a, Supplementary Fig. 23). Immunofluorescence of MP marker MyoD and proliferation marker bromodeoxyuridine (Brdu), revealed significantly more proliferating MPs in p107KO tibialis anterior (TA) muscle compared to wild type regenerating muscle, 2 days post cardiotoxin injury (Fig. 5a).

We confirmed that the proliferative differences in the p107 genetically deleted mice compared to wild type littermates were cell autonomous by considering control and p107KO c2MPs. By counting cells over several days and performing flow cytometry cell cycle analysis, we found p107KO c2MPs had almost twice the cell cycle rate, with significantly increased cell numbers in S-phase compared to control c2MPs (Fig. 5b, c, Supplementary Fig. 8), as had been previously shown with p107KO prMPs[40]. Contrarily, Sirt1KO cells that had increased levels of p107 in the mitochondria (Fig. 4b) exhibited significantly decelerated cell cycle progression with significantly fewer cells in S-phase compared to controls (Fig. 5c, Supplementary Fig. 8).

We assessed if the mitochondrial function of p107 was essential to the cell cycle rate reduction. First, we restored p107 mitochondrial localization in p107KO cells to determine if we could decrease the cell cycle rate. This was accomplished by overexpressing p107fl or p107 containing a mitochondrial localization sequence (p107mls) (Fig. 5d, Supplementary Fig. 24a). Restoring p107 to the genetically deleted cells significantly inhibited cell cycle progression by blocking cells in G1 phase of the cell cycle,

which corresponded to reduced mitochondrial gene expression (Fig. 5e, Supplementary Fig. 24b, 25). Second, we analyzed the cell cycle when endogenous p107 is significantly expressed in the mitochondria by growing cells in stripped media containing 25 mM compared to 5.5 mM glucose (Fig. 3a). In concordance with p107 mitochondrial localization, we found that S-phase was significantly reduced for cells grown only in 25 mM glucose compared to 5.5 mM (Fig. 5f, Supplementary Fig. 26a). Contrarily, S-phase was significantly increased for cells grown in complete media containing 25 mM glucose compared to 5.5 mM (Supplementary Fig. 26b). These findings suggest that p107 might direct the cell cycle rate through the management of Oxphos ATP generation.

This possibility was tested by using live-cell metabolic imaging with Seahorse. This showed that the augmented cell cycle rate in p107KO cells compared to controls corresponded to a significantly higher rate of ATP generation from Oxphos and glycolysis, including an increase in the mitochondrial to glycolytic ATP production rate ratio (Fig. 5g). In contrast, live cell metabolic imaging by Seahorse showed that p107KO c2MPs over-expressing p107fl or p107mls resulted in significant attenuation of ATP generation capacity (Supplementary Fig. 27).

We next appraised if modulation of p107 mitochondrial function to control ATP production might influence MP proliferation. For this, the cytoplasmic $NAD^+$/NADH ratio was manipulated with the addition of ox (Fig. 3j). As ox inhibits Ldha, it results in glycolytic flux relying almost entirely on $NAD^+$ generated by Oxphos. It also provides increased NADH, which augmented p107 mitochondrial localization and decreased mtDNA gene expression (Figs. 3k, l, 5h). However, ox did not affect mitochondrial gene expression in p107KO cells (Fig. 5h).

Live cell metabolic imaging of c2MPs demonstrated that the reduction in mitochondrial gene expression with ox was associated with a significant decrease in ATP generation from Oxphos as well as glycolysis (Fig. 5i, Supplementary Fig. 28). The reduced ATP production was coupled to a significant decrease in cell growth and proliferation rate (Fig. 5j, k) as well as the cell cycle rate and S-phase (Fig. 5l, Supplementary Table 3). Ox treatment of p107KO c2MPs did not alter cell cycle rate (Fig. 5j–l, Supplementary Table 3). However, when Oxphos capacity was inhibited in p107KO cells with ETC complex 1 inhibitor

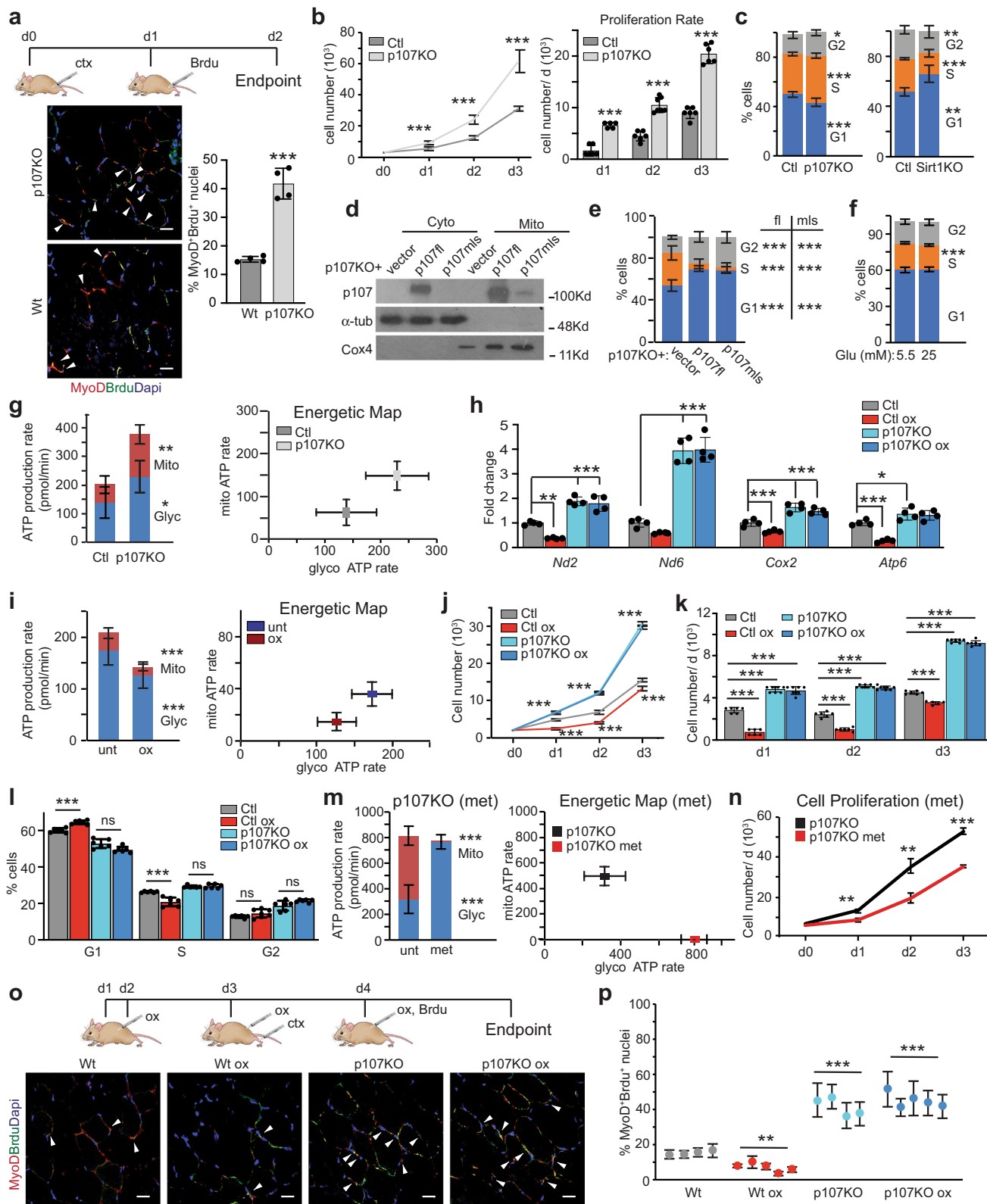

metformin (Fig. 5m), the cell cycle and proliferation rate were significantly reduced (Fig. 5n, Supplementary Fig. 29). Taken together, these results suggest that Oxphos regulation by p107 controls c2MP proliferative rates.

We confirmed these findings in vivo by treatment of mice with ox during the regeneration process. Wild type mice treated with ox showed significantly fewer proliferating MyoD[+]Brdu[+] MPs

compared to control untreated mice (Fig. 5o, p, Supplementary Fig. 30). In contrast, the muscles from p107KO mice treated or untreated with ox showed no difference in proliferating MPs (Fig. 5o, p, Supplementary Fig. 30).

Collectively, these results suggest that p107 mitochondrial function controls cell cycle rate by regulating ETC complex formation in a Sirt1 dependent manner (Fig. 6).

**Fig. 5 p107 directs the cell cycle rate through management of Oxphos generation. a** Time course of treatments for wild type (Wt) and p107KO mice injected with cardiotoxin (ctx) and bromodeoxyuridine (Brdu). Representative confocal immunofluorescence merge image of MyoD, Brdu and Dapi and graph depicting the percentage of proliferating myogenic progenitors from tibialis anterior muscle at day 2 post injury. Data are presented as mean values ± SD (n = 4 independent animals). Two-tailed unpaired Student's t test, ***p < 0.001. Arrows denote Brdu and MyoD positive nuclei. (scale bar 20 μm). **b** Growth curve and proliferation rate during 3 days for control (Ctl) and p107KO c2MPs. Data are presented as mean values ± SD (n = 6 Ctl and 8 p107KO biologically independent samples). Two-tailed unpaired Student's t test for cell number, ***p < 0.001. **c** Cell cycle analysis by flow cytometry for Ctl, p107KO and Sirt1KO c2MPs. Data are presented as mean values ± SD (n = 8 Ctl and 9 p107KO, and 8 Ctl and 13 Sirt1KO biologically independent samples). Two-tailed unpaired Student's t test; Ctl and p107KO, *p = 0.0154 G2, ***p < 0.001 G1, S, and Ctl and Sirt1KO **p = 0.0021 G1, 0.0012 G2 ***p < 0.001 S. **d** Representative Western blot of cytoplasmic (Cyto) and mitochondria (Mito) fractions of p107KO c2MPs that were transfected with empty vector alone or with full length p107 (p107fl) or mitochondria localized p107 (p107mls). **e** Cell cycle analysis by flow cytometry for cells treated as in (**d**) (n = 8 vector, 9 p107fl and 9 p107mls biologically independent samples) and (**f**) cells grown in 5.5 or 25 mM glucose (Glu) in stripped media (n = 6 biologically independent samples). Two-tailed unpaired Student's t test, ***p < 0.001. **g** Live cell metabolic analysis of Ctl and p107KO c2MPs for ATP production rate from mitochondria (Mito) and glycolysis (Glyc) and energetic map for ATP rate. Data are presented as mean values ± SD (n = 4 Ctl and 5 p107KO biologically independent samples). Two-tailed unpaired Student's t test, *p = 0.0390, **p = 0.0032. **h** RT-qPCR of Nd2, Nd6, Cox2, and Atp6 for Ctl and p107KO c2MPs in the presence or absence of 2.5 mM oxamate (ox). Data are presented as mean values ± SD (n = 4 biologically independent samples). Two-way Anova with post hoc Tukey Nd2 **p = 0.0027 (Ctl ox), ***p < 0.001 (p107KO), (p107KO ox); Nd6 p = 0.0573 (Ctl ox), ***p < 0.001 (p107KO), (p107KO ox); Cox2 ***p < 0.001 (Ctl ox), (p107KO) (p107KO ox), and Atp6 ***p < 0.001 (Ctl ox), *p = 0.0478 (p107KO), p = 0.0595 (p107KO ox). **i** Live cell metabolic analysis of c2MPs untreated (unt) or treated with ox for ATP production rate, from Mito and Glyc and energetic map. Data are presented as mean values ± SD (n = 11 unt and 8 ox biologically independent samples). Two-tailed unpaired Student's t test, ***p < 0.001. **j** Growth curve, (**k**) proliferation rate, and (**l**) Cell cycle analysis, for Ctl and p107KO c2MPs untreated or treated with ox. Data are presented as mean values ± SD (n = 6 Ctl, Ctl ox and 8 p107KO, p107KO ox biologically independent samples). Two-way Anova with post hoc Tukey***p < 0.001, ns (non-significant). For detailed statistics see Supplementary Table 3. **m** Live cell metabolic analysis of p107KO c2MPs untreated or treated with metformin (met) for ATP production rate from Mito and Glyc and energetic map for ATP rate. Data are presented as mean values ± SD (n = 12 unt and 13 met treated biologically independent samples). Two-tailed unpaired Student's t test, ***p < 0.001. **n** Growth curve for p107KO c2MPs untreated or treated with met. Data are presented as mean values ± SD (n = 4 biologically independent samples). Two-tailed unpaired Student's t test, **p = 0.0027 (d1), 0.0037 (d2), ***p < 0.001 (d3). **o** Time course treatments for Wt and p107KO mice injected with ox, ctx and Brdu and representative confocal immunofluorescence merge of MyoD, Brdu and Dapi (scale bar 20 μm). Arrows denote Brdu and MyoD positive nuclei and **p** graph depicting the percentage of proliferating MPs from tibialis anterior muscle at day 2 post injury. Data are presented as mean values ± SD (n = 4 for wt, p107KO and n = 5 for Wt ox, p107KO ox independent animals). Two-way Anova with post hoc Tukey, Wt ox **p = 0.0048, ***p < 0.001. Source data are provided as a Source Data file.

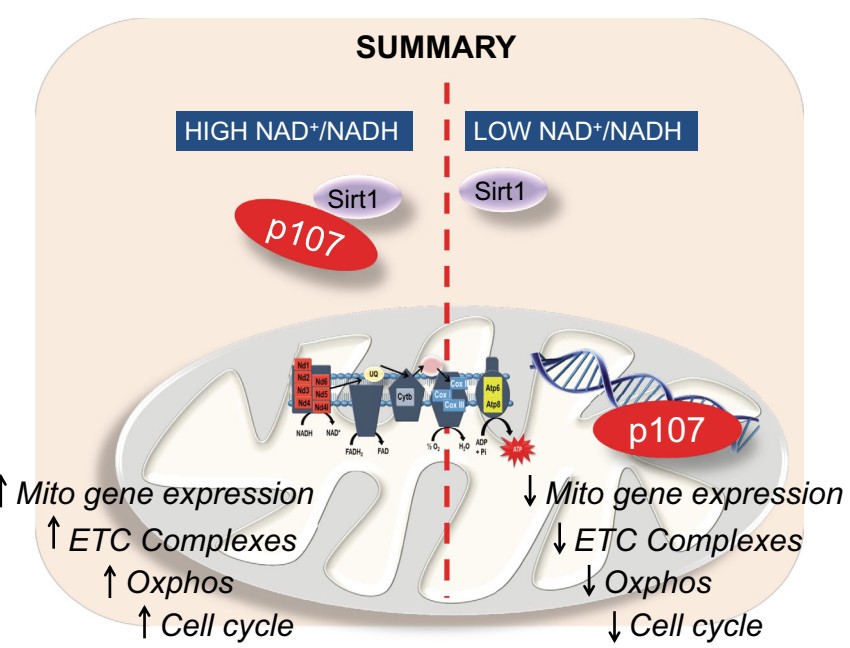

**Fig. 6 Schematic for p107 regulating muscle progenitor (MP) proliferation based on Sirt1 activity.** When the cytoplasmic NAD+/NADH ratio is high, Sirt1 is active interacting with p107 preventing its mitochondrial localization. This causes de-repression of the mitochondrial-encoded genes, thus enhancing Oxphos generation that increases MP proliferation rate. Contrarily, with low NAD+/NADH ratio, Sirt1 is inactive and p107 is free to relocate to the mitochondria, where it interacts with the mitochondrial DNA to repress its gene expression. This causes downregulation of mitochondrial Oxphos that decelerates MP cell cycle progression and proliferation rate.

## Discussion

Our results support an unanticipated mitochondrial function for p107 as a regulator of ATP generation. We used a series of complimentary approaches that underscore p107 operation within the mitochondria as a transcriptional co-repressor.

Biochemical mitochondrial fractionation showed that it is located in the matrix and interacts with the mtDNA to repress gene expression. By controlling gene expression in this manner, we found that p107 directly impacted the Oxphos capacity of MP cells and restricted their proliferation rate. Accordingly, these

findings showcase that p107 arbitrates cell cycle rate through metabolic control. Indeed, p107 mitochondrial function is controlled by the cellular redox status and in particular the energy sensor Sirt1, an $NAD^+$ dependent deacetylase. Intriguingly, as p107 is almost always shown to be expressed only in proliferating cells, the Sirt1-p107 pathway in MPs is possibly a universal cell mechanism utilized during division.

For many years, assigning a functional role for p107 has been ambiguous, unlike family member Rb that is a bona fide G1 phase restriction checkpoint factor[31]. In this study, we found that p107 acts as a metabolic checkpoint molecule for cellular energy status, placing it within the pantheon of other well-known nutrient sensing cell growth checkpoint molecules[41]. It underscores an unanticipated mitochondrial role for p107 in controlling myogenic progenitor cell cycle through regulation of mitochondrial ATP generation. This function might be exclusive to p107 during cell proliferation of myogenic progenitors, as for the other family members, Rb is not expressed in the mitochondria and p130 is not expressed at all. Though, Rb has been shown to be located at the mitochondria on the outer mitochondrial membrane of IMR90 human lung cells, it is not found in the matrix or inner membrane where mtDNA and ETC complexes are located[42]. At the outer mitochondrial membrane, Rb is thought to directly interact with the apoptosis regulator BAX to modulate apoptosis[42], not like p107, which is shown to influence mitochondrial-encoded gene expression. Moreover, p107 downregulation and genetic deletion is always associated with increased Oxphos[2,25,26,29,43], whereas Rb loss and reduced activity are attendant with both increased and decreased Oxphos that might be cell type or context dependent[44–49].

During cell division, it is unclear how glycolysis and Oxphos collectively operate to regulate the yield of ATP. Our data show a major rationale for the ubiquitous expression of p107 in proliferating cells as part of a mechanism involved in this interplay. Our results highlight that p107 influences ATP production in the mitochondria based on the energy generated in the cytoplasm, in a Sirt1 dependent manner. The lowering of the $NAD^+$/NADH ratio by a higher glycolytic flux was concomitant with a decreased capacity to generate ATP from Oxphos. We believe this is in part due to increasing levels of p107 interacting at the mtDNA where it represses gene expression to limit ETC complex formation. At this time, it is not known how p107 is shuttled across the outer and inner mitochondrial membranes. It might be a result of post-translational modification by acetylation and/or phosphorylation.

Regulation of mitochondrial gene expression is poorly understood relative to control of nuclear gene transcription. The involvement of known regulators of gene expression in the nucleus to play a role in governing mtDNA gene expression via actual localization in the mitochondria has been shown for several factors[23]. It is currently not definitive if p107 influences mitochondrial encoded gene expression and/or mitochondrial DNA copy number by interacting within the regulatory D-loop region or outside at other region(s) along the mtDNA, as has been shown for other regulators[23,50,51]. Moreover, it is not clear if the mitochondrial transcriptional regulators, including p107, operate in a cell or physiological specific context. However, our findings link a nuclear co-transcriptional repressor involved in directly repressing mitochondrial transcription to changes in metabolism and cell cycle.

The role of p107 in repressing mtDNA transcription might ensure that ATP levels remain compatible with the demands of the proliferating cells and might also help steer the TCA cycle away from producing reducing equivalents in favor of biomass production[52]. Notably, p107 is found to relocate from the cytoplasm to nucleus at late G1 or G1/S phase[53–57]. This is when a

glycolytic to Oxphos switch occurs and when it silences Ldha and Pdk2 gene expression on nuclear promoters to further increase Oxphos capacity as shown in adipocyte progenitors[2]. Thus, we propose that p107 might dynamically modulate cell metabolism during the cell cycle through promoter repression in two different organelles. Through inhibiting Oxphos in G1, dependent on the nutrient load, and increasing Oxphos in S phase, p107 is able to exert dual influences on cell cycle progression.

The high rate of ATP generated by Oxphos and glycolysis in p107KO MPs can explain the accelerated cell cycle rate. The higher proliferative rate is also characteristic of p107KO murine embryonic fibroblasts and primary MPs[40] and was evidenced in the significantly greater number of proliferating MPs within regenerating p107KO skeletal muscle. Conversely, if p107 function is dysregulated and forced to remain in the mitochondria, the reduced ATP generation capacity alarmingly decreases cell cycle efficiency. Based on these results we speculate that the decreased proliferative capacity of p107 overexpressing cells found in some reports might be attributed to a mitochondrial role. In this case, repressing mitochondrial gene expression, which is required for G1 cell cycle progression. Indeed, overexpression of p107 can inhibit cell proliferation and arrest cell cycle at G1 in many cell types[31,58,59] and delay the G1 to S phase entry in rat fibroblast cell lines[56].

Our results demonstrate that increased Oxphos is linked to an increase in cell cycle rate in p107KO compared to control cells that might be due to an enhanced supply of free $NAD^{+\,7,8}$. Recently, it was shown that promoting pyruvate oxidation by PDK inhibition, reduced the $NAD^+$/NADH ratio concomitantly with decreased cellular proliferation and ATP generation[7]. Also, we showed that increasing pyruvate oxidation with oxamate, DCA or Ldha KD, reduced the $NAD^+$/NADH ratio, Oxphos potential and cell cycle rate. Importantly, we found that these effects were tied to p107 function in the mitochondria that decreased mitochondrial gene expression. We believe that in the p107KO cells, NADH is more effectively oxidized increasing ATP generation from Oxphos that increases $NAD^+$ accessibility promoting a faster cell cycle. Furthermore, p107KO cells are not affected by NADH overload to the mitochondria, with increased pyruvate oxidation by the addition of oxamate, DCA or Ldha KD. We suppose this is due to a greater capacity for NADH oxidation by Oxphos in p107KO cells compared to controls. Indeed, when we blocked the ETC cycle with metformin in p107KO cells we reduced the cell cycle rate, thus relating the importance of Oxphos potential to the cell cycle.

Our data strongly support the concept that the $NAD^+$/NADH ratio controls p107 function through Sirt1, which is activated by $NAD^+$. Moreover, we and others have shown that Sirt1 regulation of SC and MP cell cycle parallels p107 mitochondrial control of proliferation. Indeed, Sirt1 has been shown to promote MP proliferation[60,61], which corresponds to p107 relocation from the mitochondria. Consistent with this idea, mice with activated Sirt1 (by caloric restriction or by $NAD^+$ repletion) exhibited greater SC self-renewal and more proliferative MPs[14,33,62,63]. Conversely, primary activated SCs isolated from Sirt1KO, as well as conditional Sirt1KO, that would increase p107 mitochondrial localization and repress mtDNA gene expression, showed significantly lowered cell cycle as measured by the incorporation of nucleotide base analogs into DNA[64,65]. Though the use of Sirt1KO cells confirmed that p107 functions through a Sirt1 dependent mechanism, it is unlikely that the functional interaction between p107 and Sirt1 is maintained during myogenic differentiation, as Sirt1 has a non-metabolic role during differentiation[33,66]. Nonetheless, these findings set the stage for future evaluation of the mitochondrial function of p107 during SC activation, self-renewal and commitment.

In summary, our findings establish that a cell cycle regulator, p107, functions as a key and fundamental component of the cellular metabolism network during cell division. Indeed, it directly manipulates the energy generation capacity of mitochondria by indirectly sensing the glycolytic energy production through Sirt1. These results provide a conceptual advance for how proliferating cells regulate energy generation through the interplay between glycolysis and Oxphos. Importantly because of the ubiquitous p107 protein expression in most dividing cells, the findings identify a potential universal cellular mechanism with immense implications for studies on cancer cell proliferation and stem cell fate decisions.

## Methods

**Cells and mice**. The C2C12 myoblast cell line (designated as c2MPs) was purchased from the American Tissue Type Culture (ATTC) and grown in Dulbecco's Modified Eagle Medium (DMEM) containing 25 mM glucose supplemented with 10% fetal bovine serum (FBS) and 1% penicillin streptomycin. Housing, husbandry, and all experimental protocols for mice used in this study were performed in accordance with the guidelines established by the York University Animal Care Committee, which is based on the guidelines of the Canadian Council on Animal Care. The animal use protocols were approved by the Animal Care Committee of York University. Wild type and p107KO mice from M. Rudnicki[40] were a mixed strain (NMRI, C57/Bl6, FVB/N) background[67]. The mice were maintained in a temperature-and humidity-controlled 12-hr light-dark cycle. Food and water were provided ad libitum. At 8-to-10-weeks of age, mice were used for derivation of prMPs and tissue immunofluorescence. The prMPs were isolated from single fibers obtained from the extensor digitorum longus muscles that were grown for 5 days before the media containing the prMPs was transferred to collagen coated fresh tissue culture plates. The prMPs were grown in Ham's F10 Nutrient Mix Media supplemented with 20% FBS, 1% penicillin streptomycin and 2.5 ng/ml bFGF (Peprotech).

For the nutrient-specific experiments, c2MPs or prMPs were treated with glucose, galactose, or glutamine (for detailed incubation time see Supplemental Materials and Methods). For drug-specific treatment, cells were treated with oxamate, dichloroacetic acid, nicotinamide, srt1720 or resveratrol (for detailed protocol see Supplemental Materials and Methods).

For Western blot, antibodies used, percentage of p107 cellular distribution, iMTS-like sequence determination, nuclear and cytoplasmic extraction, growth curve, and proliferation rate, see Supplemental Materials and Methods.

**Ldha knockdown**. For generating Ldha knockdown cells, Lipofectamine 2000 (ThermoFisher Scientific) was used to transfect c2MPs with mouse Ldha siRNA ON-TARGETplus Smart pool (cat #L-043884-00-0005, Dharmacon) or mouse control siRNA ON-TARGETplus control pool (cat #D-001810-10-05) according to the manufacturer's protocol. For each transfection, 50 nM of control or Ldha siRNA, or Lipofectamine 2000 were dissolved in two separate tubes with OptiMEM (ThermoFisher Scientific) and incubated for 5 min at room temperature. The tubes were then mixed and incubated for 20 min at room temperature before being added dropwise to the cells and incubated overnight at 37 °C. The following day, the transfection media was changed to 25 mM glucose DMEM with 10% serum. Cells were harvested 48 h post-transfection for assessments.

**Cloning**. The p107mls expression plasmid that expresses p107 only in the mitochondria was made by cloning full-length p107 into the pCMV6-OCT-HA-eGFP expression plasmid vector[68] that contains a mitochondrial localization signal. We used the following forward 5′-CACCAATTGATGTTCGAGGACAAGCCCCAC-3′ and reverse 5′-CACAAGCTTTTAATGATTTGCTCTTTCACT-3′ primer sets that contain the restriction sites Mfe1 and HindIII, respectively, to amplify full a length p107 insert from a p107 Ha tagged plasmid[69]. The restriction enzyme digested full-length p107 insert was then ligated to an EcoRI/HindIII digest of pCMV6-OCT-HA-eGFP, which removed the HA-eGFP sequences but retained the n-terminal mitochondrial localization signal (OCT). The calcium chloride method was used for transfections (see Supplemental Materials and Methods).

For overexpression studies, at least 4 different p107KO c2MPs were transfected as above with GFP mitochondrial localization empty vector pCMV6-OCT-HA-eGFP[68], p107fl expressing full-length p107 tagged HA, and p107mls expressing full-length p107, which is directed to the mitochondria. For Sirt1 overexpression experiments, c2MPs were transfected with p107fl alone or with full length (Sirt1fl) or dominant-negative (Sirt1dn) Sirt1[70].

**p107KO and SirtKO cell lines derivation**. Crispr/Cas9 was used to generate p107 and Sirt1 genetically deleted c2MP (p107KO and Sirt1KO) cell lines. For p107KO c2MPs, C2C12 cells were simultaneously transfected with 3 pLenti-U6-sgRNA-SFFV-Cas9-2A-Puro plasmids each containing a different sgRNA to target

p107 sequences 110 CGTGAAGTCATCCAGGGCTT, 156 GGGGAGAAGTTAT ACACTGGC and 350 AGTTTCGTGAGCGGATAGAA (Applied Biological Materials), and for Sirt1KO with 2 Double Nickase plasmids each containing a different sgRNA to target sequences 148 CGGACGAGCCGCTCCGCAAG and 110 CCATGGCGGCCGCCGCGGAA (Santa Cruz Biotechnology). For control cells, C2C12 cells were transfected by empty pLenti-U6-sgRNA-SFFV-Cas9-2A-Puro (Applied Biological Materials). See Supplemental Materials and Methods for detailed protocol.

**Cardiotoxin-induced muscle regeneration**. Three-month-old anesthetized wild type and p107KO mice were injected intramuscularly in the tibialis anterior (TA) muscle with 40μl of cardiotoxin (ctx) Latoxan (Sigma) that was prepared by dissolving in water to a final concentration of 10μM. A day after ctx injury, bromo-deoxyuridine (Brdu) at 100 mg/kg was injected intraperitonially and TA muscles were collected on day 2 post ctx injection. Mice were also untreated or treated with 750 mg/kg ox for four consecutive days, with ctx on the third day and Brdu on the fourth day, before the TA muscles were dissected on the fifth day for freezing.

**Mitochondrial isolation**. Cells were washed in PBS, pelleted, dissolved in 5 times the packed volume with isolation buffer (0.25 M Sucrose, 0.1% BSA, 0.2 mM EDTA, 10 mM HEPES with 1 mg/ml of each pepstatin, leupeptin and aprotinin protease inhibitors), and homogenized in Dounce homogenizer on ice. The homogenate was centrifuged at 1000 g at 4 °C for 10 min. The supernatant was then centrifuged at 14000 g for 15 min at 4 °C and the resulting supernatant was saved as "cytosolic fraction". The pellet representing the "mitochondrial fraction" was washed twice and dissolved in isolation buffer. The mitochondria were lysed by repeated freeze-thaw cycles on dry ice. Mitochondrial fractions were isolated using a hypotonic osmotic shock approach[37] (detailed protocol in Supplemental Materials and Methods).

**Mitochondrial and nuclear DNA content**. To obtain the relative mtDNA copy number, c2MPs or p107KO cells grown on a 6 cm tissue culture plate were untreated or treated with 1 mM 5-aminoimidazole-4-carboxamide-1-β-D-ribofuranoside (AICAR) (Toronto Research Chemicals) for 24 h in the presence of 5.5 mM or 25 mM glucose, with or without 10 mM Nam or 1μM srt1720 (detailed protocol in Supplemental Materials and Methods).

**Co-immunoprecipitation assay**. For Immunoprecipitation (IP), protein lysates were pre-cleared with 50μl protein A/G plus agarose beads (Santa Cruz Biotechnology) by rocking at 4 °C for an hour. Fresh protein A/G agarose beads along with 5μg of p107-C18, Sirt1-B7, or IgG-D7 antibody were added to the pre-cleared supernatant and rocked overnight at 4 °C. The next day the pellets were washed 3 times with wash buffer (50 mM HEPES pH 7.0, 250 mM NaCl and 0.1% Np-40) and loaded onto polyacrylamide gels and Western blotted for p107-SD9 or Sirt1-D1D7. Inputs represent 10% of lysates that were immunoprecipitated.

For IP of cytoplasmic fractions, 40ul of PureProteome protein A/G mix magnetic beads (EMD Millipore Corp) were used and incubated with 10ug of antibody; p107-SD9, Sirt1-B7 or IgG-D7 (Santa Cruz Biotechnology) at 4 °C for 30 min while rocking. After, cytosolic lysates were added for incubation overnight at 4 °C while rocking. The following day the lysates were washed, loaded onto SDS-PAGE gels and Western blotted for p107 (13354-1-AP) or Sirt1 (D1D7). Inputs represent 10% of lysates that were immunoprecipitated.

**qPCR analysis**. qPCR experiments were performed according to the MIQE (Minimum Information for Publication of Quantitative Real-Time PCR Experiments) guidelines[71] (see Supplemental Materials and Methods) and for primer sets used (Supplemental Table 4).

**Quantitative chromatin immunoprecipitation assay (qChIP)**. qChIP was performed according to De Souza et al[29], (detailed protocol in Supplemental Materials and Methods). Fold enrichment was determined by amplifying isolated DNA fragments using the D-loop primer sets (Supplemental Table 4) analyzed using the Ct method and normalized to IgG Ct values.

**NAD⁺/NADH and ATP generation assays**. $NAD^+$/NADH assay was performed as previously published[2], view detailed protocol in Supplemental materials and methods. ATP production capacity of isolated mitochondria was measured using an ATP determination kit (ThermoFisher Scientific) as per the manufacturer's instructions (Supplemental Materials and Methods). ATP production for each sample was normalized to total protein using Bradford assay kit (BioBasic).

**Immunocytochemistry and confocal imaging**. Cells were washed in PBS, fixed for 5 min with 95% methanol and permeabilized for 30 min at 4 °C with blocking buffer (3% BSA and 0.1% saponin in PBS). Then incubated with primary antibodies followed by secondary antibodies 1 h each, with intermittent washing. Finally, cells were incubated in 4′,6-diamidino-2-phenylindole (Dapi) and confocal

images and Z-stacks were obtained using the Axio Observer.Z1 microscope with alpha Plan-Apochromat 63x/Oil DIC (UV) M27 (Zeiss) using Axiocam MR R3. (See Supplemental Materials and Methods for detail) For antigen retrieval protocol and immunocytochemistry with MitoTracker red see Supplemental Materials and Methods).

**Mitochondrial length and area measurement.** Live cell imaging was done using the Axio Observer.Z1 (Zeiss) microscope with alpha plan-apochromat 40x/Oil DIC (UV) M27 (Zeiss) in an environment chamber (5% $CO_2$; 37 °C) using Axiocam MR R3 (Zeiss). Cells were stained with MitoView red (Biotium) and the mitochondrial length and area were measured by using Image J software (detailed protocol in Supplemental Materials and Methods).

**Immunohistochemistry.** Frozen muscle tissue samples were fixed with 4% paraformaldehyde for 15 min and for antigen retrieval 2 N HCl was added for 20 min followed by 40 mM sodium citrate. After washing in PBS, muscle sections were blocked in blocking buffer (5% goat serum, 0.1% Triton X in PBS) for 30 min. The sections were then incubated with primary antibodies followed by secondary antibodies with intermittent washes. Finally, Dapi was added and the sections were imaged using confocal microscopy with the Axio Observer.Z1 microscope with alpha Plan-Apochromat 40x/Oil DIC (UV) M27 (Zeiss). See Supplemental Materials and Methods for detail.

**Flow cytometry.** For cell cycle analysis 50000, Ctl and p107KO cells were treated or untreated with 2.5 mM ox for 40hrs or c2MPs were grown in 5.5 mM or 25 mM glucose in stripped or complete media for 20 h or Ctl and Sirt1KO cells were grown in 5.5 mM glucose (which normally activate Sirt1) for 20 h or p107KO cells 24 h post-transfection with pCMV6-OCT-HA-eGFP alone or together with p107fl or p107mls or p107KO cells untreated or treated with 5 mM metformin for 24 h or c2MPs synchronized at G1 and G2 phases of cell cycle were used.

Following fixation and washing, the cells were incubated in 50µg/ml propidium iodide (ThermoFisher Scientific) and 25µg/ml RNAse (ThermoFisher Scientific) before loading on the Attune Nxt Flow Cytometer (Thermo Fisher Scientific). See Supplementary Materials and Methods for detailed protocol. Cells were synchronized before assessment at G1 and G2 phases of cell cycle. For G1, cells were washed twice with PBS and grown in DMEM containing 1% FBS and 1% penicillin streptomycin for 72 h. For G2, c2MPs were washed twice with PBS, refed and treated with 50 ng/ml Nocodazole (Selleckchem Chemicals) for 18 h. Cells were analyzed for PI using ModFit LT (8.0) (Supplementary Figs. 8, 25). For detailed protocol see Supplemental Materials and Methods.

**Live cell ATP analysis (Seahorse).** 3000 cells were seeded in DMEM (Wisent) containing 10% FBS and 1% penicillin streptomycin on microplates (Agilent Technologies) and treated according to the required experiment. The cells that were used were control or p107KO cells or p107KO cells transfected with pCMV6-OCT-HA-eGFP alone or together with p107fl or p107mls or p107KO cells untreated or treated with 5 mM metformin or c2MPs and Sirt1KO c2MPs untreated or treated with 1 µM srt1720. For analysis, cells were washed in XF assay media supplemented with 10 mM glucose, 1 mM pyruvate and 2 mM glutamine (Agilent Technologies) and assessed using the Seahorse XF real-time ATP rate assay kit (Agilent Technologies) on a Seahorse XFe96 extracellular flux analyzer (Agilent Technologies), with the addition of 1.5 µM of oligomycin and 0.5 µM rotenone + antimycin A as per the manufacturer's direction. The energy flux data in real time was determined using Wave 2.6 software (Agilent Technologies).

**Statistics and reproducibility.** No statistical method was used to pre-determine sample size. All experiments were performed with at least three biological replicates as indicated in the figure legends, and results are presented as the mean ± standard deviation (SD). The immunoblot (Fig. 1a, c, d, n, 2b, g, 4b, e, k, 5d and Supplementary Figs. 5a, b, 13a, 18a, 19a), co-immunoprecipitation (Fig. 4a and Supplementary Fig. 15) and immunostaining (Fig. 1e, g, h, j, l, and Supplementary Figs. 1, 4a, c, d, f, 6, 20, 21, 23, 24a, 30) experiments have been performed at least three independent times with similar results. The experimental design incorporated user blinding when possible. Statistical analysis was performed using GraphPad Prism. Statistical comparisons between groups were made using two-tailed unpaired Student's $t$ test, or as appropriate, one-way or two-way analysis of variance (ANOVA) with a criterion of $p < 0.05$. All significant differences for ANOVA testing were evaluated using a Tukey post hoc test. Results were considered statistically significant when $p < 0.05$. The level of significance is indicated as follows: *$p < 0.05$, **$p < 0.01$, ***$p < 0.001$.

Figures 1k, 5a, 5o, and 6 were created using BioRender (www.biorender.com), Adobe Illustrator (v25.3.1) and Microsoft PowerPoint.

**Reporting summary.** Further information on research design is available in the Nature Research Reporting Summary linked to this article.

## Data availability
The raw images for the immunoblots and datasets for the graphs are provided in the Source Data file. Source data are provided with this paper.

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

## Acknowledgements

The authors thank Drs. Mireille Khacho and Tara Haas for critical insights and Dr. Haas for careful reading of the manuscript. Drs. Magdalena Jaklewicz and Geetika Phukan and Mayoorey Murigathasan for technical support. A.S. is a recipient of a Natural Sciences and Engineering Research Council of Canada research grant (RGPIN-2018-05937).

## Author contributions

A.S. planned and managed the research activity, wrote the original draft and acquired financial support for the project. D.B., V.S., and A.S. critically appraised and reviewed the draft, made the figures, formulated the ideas, developed and designed the methodology, validated the research outputs. D.B., A.S., V.S. and O.O. conducted the experiments and formally analyzed the data.

## Competing interests

The authors declare no competing interests.
