## [Peer Review File · Nature Communications]

p107 mediated mitochondrial function controls muscle stem cell proliferative fatesREVIEWER COMMENTS

Reviewer #1 (Remarks to the Author):

In this study, Battacharya et al. identify a novel role for the RB family member p107 in regulating mitochondrial ATP generation. They found that p107 binds to mitochondrial DNA in myogenic progenitor (MP) cells and represses the expression of mitochondrially-encoded genes that function in the electron transport chain, leading to reduced ATP production in mitochondria. Intriguingly, the function of p107 at mitochondria is regulated by the metabolic state of the cell. A high NAD⁺/NADH ratio activates the deacetylase Sirt1, which in turn binds p107 and prevents p107 from entering mitochondria, ultimately resulting in increased ATP production. Loss of p107 through CRISPR/Cas9-mediated gene knockout leads to similar increases in ATP production, as well as faster cell proliferation. Importantly, p107 targeted to mitochondria is capable of arresting cells in G1 to a similar extent as wild-type p107, suggesting that p107's control of the cell cycle is through this metabolic regulation in MP cells. Finally, the authors use a mouse model of muscle injury to demonstrate that proliferation of MPs following injury decreases with decreasing NAD⁺/NADH ratio, and that this phenotype is dependent on the presence of p107.

The authors provide compelling evidence that p107 plays a critical role in regulating ATP production from mitochondria. By using p107 KO cells as a control throughout the study, they demonstrate that the link between upstream metabolic changes and mitochondrial function depends on p107. This is exciting work, as it hints at a mechanism to coordinate cell proliferation and cell metabolism. As the authors point out, one could imagine this mechanism operating in multiple, if not all, cell types. These data and a role for p107 in inhibiting mitochondrial oxphos could help reduce ROS production in cancer cells and may explain why p107 is rarely mutated in human cancer. However, this study could be improved by strengthening the link between the metabolic and proliferation phenotypes, considering what is already known about p107, and by further exploring the physiological relevance of this mechanism of metabolic regulation. Specifically, the following points should be addressed.

1) An important control for all the figures comparing control and p107 knockout cells (e.g. Figures 1 and 2), especially based on the cell cycle differences in Figure 5, is to determine whether the cell cycle of these control and knockout cells is the same. Otherwise, it becomes difficult to interpret the data, especially since the authors propose that p107 plays different metabolic roles at different stages of cell cycle progression. The authors should show cell cycle analyses of the cells they study, for example BrdU incorporation or western blot for cell cycle proteins whose levels change during cell cycle progression.

2) A key observation made by the authors is that of the mitochondrial localization of p107 and its ability to repress the expression of metabolic genes there.

First, some additional controls/explanations are needed. It is surprising in Figure 1B to detect so much p107 in growth-arrested cells (p107 should only be transcribed in cycling cells). How were the cells growth-arrested in the experiments shown? Does p107 localization to the mitochondria change throughout a normal cell cycle? It is also surprising in the immunofluorescence images that so little p107 is detected in the nucleus compared to the western blot data, do the authors have any explanation for this? What does the immunofluorescence signal look like in quiescent cells where the western blot suggests no presence in mitochondria but presence in cells?

Second, p107 relies on transcription factors such as E2F to bind to DNA and transcriptional repressor complexes to silence genes (e.g. DREAM complex). How do the authors envision p107's role as a repressor of gene expression in mitochondria? It may be difficult to identify these mechanisms but the authors should at least determine if E2F4/5 and/or DREAM members are localized in the mitochondria or if a mutant p107 that does not bind E2F (e.g. PMID: 7799940 but also in more recent studies) still represses mitochondrial genes

Is this mitochondrial function specific to p107, or do RB and/or p130 play similar roles? Hilgendorf and colleagues identified RB at the mitochondria but proposed a different role (PMID: 23618872). This study should have been cited and discussed in the context of these new findings. A simple first experiment may be to express a form of RB or p130 that is directed to the mitochondria (and controls) and analyze DNA binding or gene expression. The strong effects observed on gene expression in Figure 2B for example would suggest that this might be a unique function of p107 (no obvious compensation by RB) but then would raise the question of the specific signals in the p107 protein that directs it to the mitochondria. Are the first 50-100 amino acids of p107 enough to form a mitochondria localization signal?

Is the binding between SIRT1 and p107 dependent on p107 phosphorylation by Cdk2/4/6 (which could be tested by expressing a form of p107 that cannot be phosphorylated on the main Cdk2/4/6 targets, or possibly by pre-treating the protein extracts with a phosphatase before co-IP). It may be more difficult to determine whether SIRT1 deacetylates p107, but this could be another possibility.

3) The regulation of p107 mitochondrial activity by metabolic pathways is very exciting.

Does the function of p107 in the nucleus (i.e. E2F binding) also change in response to changes in NAD⁺/NADH ratio and/or Sirt1 activity?

Can the increased cell proliferation observed in p107 KO cells be rescued by decreasing ETC function downstream of p107 (e.g. through complex I inhibition with metformin)?

Figure 3A: how long were cells grown in the various concentrations of glucose and could the authors show total levels of p107? Are the differences due to changes in the cell cycle?

Loss or inactivation of Sirt1 seems to lead to an increase in p107 (Fig. 4B,E). Is this increase at the mRNA or protein level? Could this be a confounding factor that makes it appear as though p107 is translocating to the mitochondria, when in fact it is just more easily detected due to increased total expression?

Figure 4M shows that resveratrol treatment increases mitochondrial gene expression even in the absence of p107, suggesting that either this drug has off-target effects, or that the link between Sirt1 activity and mitochondrial output occurs through additional factors other than p107. The authors should address this in the text.

The Seahorse assays in Fig. 5F and Supp. Fig. 13B show that the presence or absence of p107 alters ATP production from both glycolysis and the mitochondria. Why is glycolytic output changing, when the author's model places p107 downstream of glycolysis? Is this an indirect effect, or is p107 influencing metabolism through multiple mechanisms? Because ATP output from both pathways is affected by changes in p107 levels, one cannot conclude that it is specifically p107's regulation of oxphos that is influencing cell proliferation.

Reviewer #2 (Remarks to the Author):

In this manuscript, the authors investigate the role of p107 in mitochondrial metabolism. p107 is a repressor of E2F transcription factors in the mitotic cell cycle. There are very few functions that are known to be specific to p107 compared to RB and p130. The authors have carefully investigated the localization of p107 in mitochondria, its binding to (mitochondrial) chromatin, gene expression, and metabolism. The interaction of p107 with SIRT1 is interesting and novel.

The authors present a lot of data and in most cases appropriate controls are included. Overall, it is an interesting story but to some degree it is very p107-centric. The authors should tone down their claims since it is very well known that p107 functions in repressing E2F (which is known to affect metabolism – see L. Faja's work). I would encourage the authors to be more cautious with the interpretation of their data and how to put it into context with what has been published before.

These are the issues, which need to be addressed before this manuscript can be considered:

1. In the introduction the authors indicate that proliferating cells need more ATP and non-proliferating cells. For a long time, I was thinking the same until I discussed this with Matthew Vander Heiden. He had made measurements and found that this was not true. Therefore, I would recommend that the authors use caution in that respect.
2. Figure 1: it seems that only a small fraction of p107 is localized to the mitochondria judging from WB (1B). In the confocal analysis, almost all p107 is localized to the mitochondrial and the signal in the nucleus is almost undetectable (1D). Somehow these results do not add up. Can the authors comment on this? BTW, why did the authors not use mitotracker as a marker for mitochondria?
3. Figure 2A: in the ChIP experiments, the y-axis is "Promoter Occupancy" which is unusual. Most scientist will use percent compared to input. Do they authors have a specific reason why they are not following convention?
4. Figure 2B-D: these results are interesting but do not prove that p107 is responsible for these changes. It is correlative. The authors should be cautious when they describe and interpret these kinds of data.
5. Figure 3A-B: is it glucose dependent or proliferation dependent? This is important since higher glucose levels could stimulate proliferation.
6. Figure 3F: this result is counterintuitive unless high glucose levels are toxic for these cells. The authors should explain this more carefully.
7. The relation of p107 and the NAD⁺/NADH ratio [probably better to calculate the NAD⁺/(NADH+NAD⁺) ratio] is a chicken and egg question. I would suggest that the authors are careful with this.

Reviewer #3 (Remarks to the Author):

The authors are motivated by their interest in the metabolic states which govern muscle stem cells proliferation and differentiation. Thus, they focus on the regulation of glycolysis versus OXPHOS. In the frame of this interest they identified p107 – a member of the RB protein family, which is here shown to localized in the mitochondria of mouse cells. Previously (2017), the same authors showed association between reduced expression of p107 and increase in OXPHOS protein levels and function in human muscle in response to exercise. This suggested a repressive activity of p107 on mitochondrial function. In the current manuscript, the results indeed demonstrate that mitochondrial localization of p107 associate with reduced mtDNA gene expression in several primary and transformed mouse cells. Therefore, one can view this as a follow-up study. Secondly, the authors suggest that the involvement of p107 in mtDNA gene expression occurs via mtDNA binding at the D-loop based on ChIP-qPCR results. They show that such reduced gene expression increases the length of the cell doubling time, and reduce mitochondrial ATP production. Then, they show evidence that the very mitochondrial localization of p107 negatively correlates with the expression of SIRT1, i.e. SIRT1 expression attenuates mitochondrial localization of p107 which leads to expression of mtDNA genes (with a continuous mitochondrial localization of p107 upon SIRT1 KO).

In general, the results in this manuscript are impressive, encompassing gene manipulation in cells (primary cell culture and immortalized cells) and in the whole organism (mouse). In my personal opinion, showing that an RB-interacting protein (p107) directly affects mitochondrial activity is very important, as it offers connection of cell cycle regulation to mitochondrial regulation. Nevertheless, there are several points that I feel should be revised to improve the manuscript:

1. Although much information is given regarding the impact of reduced mtDNA gene expression in

response to mitochondrial localization of p107, I missed discussion and experiments which relate to the mechanism by which such reduced expression occurs. In other words, for the past five decades much data accumulated regarding the core elements and factors that govern mtDNA transcription. These factors are not mentioned at all in the current manuscript, namely the RNA polymerase of the mitochondria - POLRMT, transcription factor A - TFAM, TFB2M, mitochondrial transcription elongation factor - TEFM and mitochondrial transcription termination factors MTERF. There are multiple papers published in this field including relatively recent reviews in good journals. Here are two major examples: Gustaffson et al Annual Rev of Biochemistry (2016); Barshad G et al Trends in Genetics, 2018. How is the impact of p107 on mtDNA gene expression relates to the mentioned factors and core mechanism of mtDNA transcription? It is essential to discuss p107 in the context of mtDNA transcriptional regulation and its relations with such regulatory factors.

2. In addition, the involvement of a known regulator of gene expression in the nucleus also in the regulation of mtDNA gene expression via actual localization in the mitochondria has been shown for several factors, such as MEF2D (She et al. JCI 2011), MOF (Chatterjee et al. Cell 2016) and NFATC (Lambertini et al The International Journal of Biochemistry & Cell Biology 64 (2015) 212–219). The claim that p107 affects mtDNA gene expression via mitochondrial localization and possibly via mtDNA binding should be argued in the context of such factors, and others (all summarized in the review paper that I already mentioned above - Barshad G et al Trends in Genetics, 2018). All these papers including this review, should be cited in the right context (introduction and discussion sections).

3. Inaccuracies: in the 2nd paragraph of the introduction the authors claim the following: "NADH is a by-product of glycolysis that might be used as a reducing reagent required for Oxphos". This point is incorrect: NADH for the OXPHOS is generated in the frame of the TCA cycle by Malate dehydrogenase. NADH from glycolysis cannot be imported into the mitochondria as there is no import machinery for NADH, yet its electrons can be indirectly imported from the cytosol to the mitochondria. The authors are asked to revise the sentence while taking into account the mentioned facts.

4. Throughout the manuscript the authors consider the impact of SIRT1 on p107, while not considering at all the other 6 members of the SIRTUIN family, especially SIRT3 which has a known mitochondrial localization and activity. The authors should relate to the entire SIRT family and justify why SIRT1 is the only one mentioned and manipulated.

5. Binding experiments of p107 to the mitochondrial genome were performed using ChIP-qPCR, focusing on the D-loop. This experiment assumes that protein binding there will affect the regulation of the mitochondrial genome. However, factors that bind the mtDNA in addition to the core regulators of mtDNA transcription (as mentioned in comment 2), may bind outside of the D-loop, yet affect mtDNA transcription (see especially the papers about MEF2D and MOF - mentioned above in comment 2). In addition, screen of ChIP-seq experiments available from ENCODE revealed mtDNA binding by additional known regulators of nuclear gene expression, which were experimentally shown to localize both in the nucleus and in the mitochondria in human cells (Blumberg et al. 2014 Genome Biol and Evol). The experiment done by the authors assumes D-loop binding by p107, which is not necessarily the case. Hence, the impact of p107 on mtDNA gene expression does not necessarily occur via D-loop binding - we will not know that until ChIP-seq experiments are performed. This point should be mentioned by the authors in the manuscript.

6. The impact of p107 on mitochondrial gene expression does not consider coordination between mtDNA and nuclear DNA-encoded OXPHOS genes, which could be easily measured by RNA-seq experiments of at least some of the tested cells. This should be performed.

7. In the bottom of page 5 the authors wrote: "The importance of p107 to mitochondrial gene expression was confirmed with p107KO c2MPs and prMPs, which both exhibited significantly increased mitochondrial encoded gene expression in the genetically deleted cells compared to their controls (Fig. 2C)." Increased expression of mtDNA-encoded genes is indeed shown, but it could be due to overall altered numbers of mitochondria in the cell, not necessarily due to regulation of transcription. This should be assessed by measurement of mtDNA copy number. With this in mind, in page 8 the authors wrote: "Moreover, the mtDNA to nuclear DNA ratio remained unchanged when grown in the different glucose concentrations, in contrast to MPs treated with the mitochondrial biogenesis activator AICAR (Fig. 3E)." This is totally confusing: mtDNA to nuclear DNA ratio measurement seems to me assessment of mtDNA copy number; why was it not

assayed in the p107 KO? The authors are asked to add this assay here as well and interpret the impact of p107 appropriately.

8. In page 7 of the manuscript the authors wrote: "Interestingly, we found that p107KO c2MPs had an elongated mitochondrial network (Fig. 2I) made up of mitochondria with significantly increased length and area (Fig. 2J)." Aside from mitochondrial fusion, such elongation could also be due to increased numbers of mitochondria (see previous comment) which under the microscope may appear as elongated structures. The authors are asked to refer to such possibility.

Minor comments

1. In the 2nd paragraph of the introduction the authors wrote: "Whereas, NAD⁺, a coenzyme in various metabolic pathways, is the oxidized form of NADH that can be made in glycolysis from the transformation of pyruvate to lactate". This sentence has no end – please re-write; also it has to be corrected with respect to the previous sentence about NADH: NAD⁺ is a cofactor of the SIRTUINS inside the mitochondria (SIRT3) - using the NAD⁺ from the TCA, and NAD⁺ from the glycolysis is used as cofactor for the rest of the Sirtuins outside of the mitochondria.

2. In page 16 the authors wrote: "Thus, the decreased proliferative capacity of p107 over expressing cells found in many reports might now be attributed to a mitochondrial role in repressing mitochondrial gene expression, which is required for G1 cell cycle progression." This statement may reflect over-interpretation of the data, as p107 could also be found in the nucleus with unknown function there.

Reviewer #4 (Remarks to the Author):

Bhattacharya et al. report that p107 regulates mitochondrial function and muscle stem cell proliferation fates.

The authors report the following:

1. p107 localizes in the mitochondria of myogenic progenitor cells:

COMMENTS:

Figure 1B: Of the total p107, what's the percentage that localizes to mitochondria?

Figure 1C: The image resolution doesn't allow to finally conclude that p107 and Cox4 colocalize. p107 mitochondrial localization should be performed with MitoTracker Red (which measure mitochondrial mass and membrane potential) and Green (which measures mitochondrial mass regardless of mitochondria activity).

2. P107 interacts at the mtDNA

COMMENTS:

Figure 2. Control experiments with known mtDNA binding proteins (TFAM, mitochondrial RNA polymerase POLRMT) should be performed.

3. NAD⁺/NADH regulates p107 mitochondrial function

COMMENTS:

Figure 3. Beside influencing NAD⁺/NADH ratio, oxamate and DCA affect other metabolites. For

instance, DCA increases acetyl-CoA. LDHA knock-down would be more appropriate and would confirm the result obtained with oxamate.

4. SIRT1 directly regulates p107 mitochondrial function

COMMENTS:

Figure 4A. It is not clear in which cell compartment p107 interacts with SIRT1.

Figure 4B. The levels of p107 are greatly increased in Sirt1KO cells. Is this a transcriptional or a protein stabilization effect? Total increase of p107KO in SIRT1KO cells complicates the interpretation of the results related mitochondrial localization of p107.

Figure 4K. Resveratrol leads to replicative stress and S phase transit and is independent of Sirt1 (Benslimane et al. *Molecular Cell* DOI:<https://doi.org/10.1016/j.molcel.2020.07.010>) and should not be employed to evaluate Sirt1 function.

Reviewer #1 (Remarks to the Author):

In this study, Battacharya et al. identify a novel role for the RB family member p107 in regulating mitochondrial ATP generation. They found that p107 binds to mitochondrial DNA in myogenic progenitor (MP) cells and represses the expression of mitochondrially-encoded genes that function in the electron transport chain, leading to reduced ATP production in mitochondria. Intriguingly, the function of p107 at mitochondria is regulated by the metabolic state of the cell. A high NAD⁺/NADH ratio activates the deacetylase Sirt1, which in turn binds p107 and prevents p107 from entering mitochondria, ultimately resulting in increased ATP production. Loss of p107 through CRISPR/Cas9-mediated gene knockout leads to similar increases in ATP production, as well as faster cell proliferation. Importantly, p107 targeted to mitochondria is capable of arresting cells in G1 to a similar extent as wild-type p107, suggesting that p107's control of the cell cycle is through this metabolic regulation in MP cells. Finally, the authors use a mouse model of muscle injury to demonstrate that proliferation of MPs following injury decreases with decreasing NAD⁺/NADH ratio, and that this phenotype is dependent on the presence of p107.

The authors provide compelling evidence that p107 plays a critical role in regulating ATP production from mitochondria. By using p107 KO cells as a control throughout the study, they demonstrate that the link between upstream metabolic changes and mitochondrial function depends on p107. This is exciting work, as it hints at a mechanism to coordinate cell proliferation and cell metabolism. As the authors point out, one could imagine this mechanism operating in multiple, if not all, cell types. These data and a role for p107 in inhibiting mitochondrial oxphos could help reduce ROS production in cancer cells and may explain why p107 is rarely mutated in human cancer. However, this study could be improved by strengthening the link between the metabolic and proliferation phenotypes, considering what is already known about p107, and by further exploring the physiological relevance of this mechanism of metabolic regulation. Specifically, the following points should be addressed.

1) An important control for all the figures comparing control and p107 knockout cells (e.g. Figures 1 and 2), especially based on the cell cycle differences in Figure 5, is to determine whether the cell cycle of these control and knockout cells is the same. Otherwise, it becomes difficult to interpret the data, especially since the authors propose that p107 plays different metabolic roles at different stages of cell cycle progression. The authors should show cell cycle analyses of the cells they study, for example BrdU incorporation or western blot for cell cycle proteins whose levels change during cell cycle progression.

We immensely thank the reviewer. Through addressing his/her important insights and recommendations, our manuscript has been greatly strengthened. These are detailed below point by point.

Initially, we had some difficulty interpreting the reviewer's comments. The reviewer's request "to determine whether the cell cycle of these control and knockout cells is the same" has been shown in Figure 5 where p107KO cells have a faster rate. We believe the reviewer's meaning was that the mitochondrial effects by the presence or absence of p107 for the control and knockout cells, respectively, might be affected by the phase of the cell cycle. In this case, we

interpreted the reviewer's concern to be that gene expression and Oxphos potential might not be linked to the absence of p107 in the knockout cells, but rather to the stage of the cell cycle, which might be different than in the control cells. His/her request to show cell cycle analyses of knockout and control cells with BrdU incorporation or western blot for cell cycle proteins would not be sufficient since p107KO cells cycle faster (**Fig. 5**). These assessments would be unable to distinguish the effect of p107 (its presence or absence) for mitochondrial function versus the potential confounding effect of the cell cycle phase. Hence, to sufficiently answer that the impact to mitochondrial function is directly affected by p107, we have eliminated the potential confounding effect of the cell cycle in comparing control and knockout cells (we believe this is the reviewer's request for this concern). This was accomplished by identically harmonizing the cell cycles of the control and p107KO cells to the G1 phase, such that the percentage of cells in any given phase of their cycle is identical for both types of cells (**new Suppl Fig. 8**). We found that the p107KO cells had significantly greater mitochondrial encoded gene expression compared to controls when both cell types were predominately in G1 phase of the cell cycle, also their cell cycle profiles matched (**new Fig 2F and new Suppl. Fig. 8**). Together these results verify that our gene expression findings (**now Fig. 2E**) between control and p107 knockout cell line and primary cells are not due to differences in the cell cycle phase. We now write beginning on page 7:

“We eliminated the potential confounding effect of the cell cycle state in comparing mitochondrial encoded gene expression in control and p107 knockout cells. This was undertaken by harmonizing their cell cycles so that the number of cells in any cell cycle phase was identical for both cell types. Cells were synchronized to the G1 phase of the cell cycle (**Suppl. Fig. 8**), a time point when p107 is present in the mitochondria (**Fig. 1N**). As with the asynchronous cells (**Fig. 2E**), we found that the p107KO cells had significantly greater mitochondrial gene expression compared to controls with identical cell cycle profiles (**Fig. 2F**).” We have updated the figure legends and materials and methods for the new data.

2.i A key observation made by the authors is that of the mitochondrial localization of p107 and its ability to repress the expression of metabolic genes there. First, some additional controls/explanations are needed. It is surprising in Figure 1B to detect so much p107 in growth-arrested cells (p107 should only be transcribed in cycling cells). How were the cells growth-arrested in the experiments shown?

Thank you for bringing this to our attention. These cells were not truly growth arrested, that is in a complete G0 state, otherwise p107 should not be expressed. The cultures were freshly contact inhibited, that is the cells were kept confluent on the tissue culture dish for 24hrs. We did not name this state properly. To better understand the context of the cell phenotype we have now changed the following statement “...but almost absent when the cells were contact inhibited in confluent growth arrested cultures.” to on page 5 “...but almost absent at the onset of contact inhibition in confluent cultures we designated as “growth arrested” (**Fig. 1B & Suppl. Fig. 1**)”

2.ii Does p107 localization to the mitochondria change throughout a normal cell cycle?

As per the reviewer's request, we analyzed the mitochondrial distribution of p107 in G1 and G2 phases of the cell cycle, achieved through serum starvation and nocodazole treatment,

respectively (**new Fig. 1M, new Fig. 1N & new Suppl. Fig. 7**). We found that p107 was localized predominately in the mitochondria and absent from the nucleus during G1 phase. In G2 phase it was present in both the mitochondria and nucleus. For these new data we now write on page 6 “**We determined p107 was localized in the mitochondria by Western blotting cellular fractions of cells that were almost entirely in the G1 or G2 phases of the cell cycle (Fig. 1M & Suppl. Fig. 7). In G1, p107 was expressed in the mitochondria and absent in the nucleus, contrary to G2 where it was expressed both in mitochondria and the nucleus (Fig. 1N). These data suggest that p107 might have an exclusive mitochondrial and not a nuclear function during the G1 phase of the cell cycle.**” We have updated the figure legends and materials and methods for the new data.

2.iii It is also surprising in the immunofluorescence images that so little p107 is detected in the nucleus compared to the western blot data, do the authors have any explanation for this?

We believe that the reason for the p107 level discrepancies in the nucleus between the Western blotting and confocal microscopic analysis has to do with differences in the approaches used. First, we used different antibodies for microscopic analysis (monoclonal anti-p107 SD9, Santa Cruz Biotech) versus Western blotting (polyclonal anti-p107 13354-1-AP, Proteintech) that recognize distinct epitopes on p107. Thus, the availability of these epitopes under the two experimental manipulations may have resulted in different detection sensitivities. For example, the fixing protocol used in confocal microscopy may not have been optimal for detection of nuclear p107.

2.iv What does the immunofluorescence signal look like in quiescent cells where the western blot suggests no presence in mitochondria but presence in cells?

As requested by the reviewer, we now provide immunofluorescence visualization of p107 in “growth arrested cells”. We used antigen retrieval to highlight more pronounced nuclear p107 expression that is represented by **new Supplemental Figure 1**. These images show that there is less p107 present in mitochondria compared to other compartments of the cells, which aligns with the Western blotting data in **Figure 1B**. These new data are referred to in the sentence on page 5: “**....but almost absent at the onset of contact inhibition in confluent cultures we designated as “growth arrested” (Fig. 1B & Suppl. Fig. 1).**”

2.v Second, p107 relies on transcription factors such as E2F to bind to DNA and transcriptional repressor complexes to silence genes (e.g. DREAM complex). How do the authors envision p107’s role as a repressor of gene expression in mitochondria? It may be difficult to identify these mechanisms but the authors should at least determine if E2F4/5 and/or DREAM members are localized in the mitochondria or if a mutant p107 that does not bind E2F (e.g. PMID: 7799940 but also in more recent studies) still represses mitochondrial genes

As per the reviewer’s request we have detailed a prospective interacting partner of p107 in the mitochondria. We assessed Tfam, a potential mtDNA initiation factor, as well as p107 nuclear

binding proteins on nuclear DNA, E2f4 and E2f5. Western blotting of c2MP proliferating mitochondria fractions reveal that there is negligible Tfam and no E2f5 protein expression (**new Fig. 2B**). However, we found that E2f4 protein is expressed in the mitochondria of proliferating c2MPs (**new Fig. 2B**). Further analysis with ChIP revealed that E2f4 interacts at the mtDNA during proliferation, and Tfam does not, but the opposite is the case in growth arrested cells (**new Fig. 2C**). The other important initiation factors, Polrmt and Tf2b2m were not ChIPed because their mouse specific antibodies capable of immunoprecipitation are not commercially available nor has their use been published for ChIP. For these new results we now write on page 7 in the Results section:

“As p107 does not directly interact with DNA, we assessed potential interacting transcription factors in the mitochondria. We evaluated potential mitochondrial role for the putative mitochondrial transcription factor Tfam, as well as the p107 interacting nuclear transcription factors E2f4 and E2f5³¹. By Western blotting, we found that E2f4 was present in proliferating mitochondrial lysates, compared to negligible levels of Tfam and the complete absence of E2f5 (**Fig. 2B**). Moreover, ChIP analysis indicated that E2f4, but not Tfam, interacted at the mtDNA of proliferating c2MPs, whereas the opposite pattern of interaction was found during growth arrest (**Fig. 2C**). Intriguingly, these results suggest that E2f4 might be a mitochondrial binding partner of p107 at the mtDNA during proliferation of c2MPs.”

The new figures and experiments required us to update figure legends and the materials and methods section.

2.vi Is this mitochondrial function specific to p107, or do RB and/or p130 play similar roles? Hilgendorf and colleagues identified RB at the mitochondria but proposed a different role (PMID: 23618872). This study should have been cited and discussed in the context of these new findings. A simple first experiment may be to express a form of RB or p130 that is directed to the mitochondria (and controls) and analyze DNA binding or gene expression.

As per the reviewer’s concern, the specificity of mitochondrial p107 function was assessed by determining if p130 and Rb protein was present in the mitochondria by Western blotting. In proliferating cells, we found that Rb was not detectable in the mitochondria and p130 was not detectable in any cell fractionl (**new Fig. 1C**). For this new finding, we now write on page 5 of the Results section: “Unlike p107, family member Rb1 (Rb) was not expressed in the mitochondria and Rb12 (p130) was not detectable in any cellular compartment in proliferating cells, (**Fig. 1C**).” The new figures and experiments required us to update figure legends and the materials and methods section

Also, as per the reviewer’s suggestion we have now discussed the findings of Hilgendorf et al and regarding Rb mitochondrial localization and role in Oxphos generation in the context of our findings. We have added to the Discussion section on page 18 the following:

“It underscores an unanticipated mitochondrial role for p107 in controlling myogenic progenitor cell cycle through regulation of mitochondrial ATP generation. This function might be exclusive to p107 during cell proliferation of myogenic progenitors, as for the other family members, Rb is not expressed in the mitochondria and p130 is not expressed at all. Though, Rb has been shown to be located at the mitochondria on the outer mitochondrial membrane of IMR90 human lung cells, it is not found in the matrix or inner membrane where mtDNA and ETC complexes are

located⁴². At the outer mitochondrial membrane, Rb is thought to directly interact with the apoptosis regulator BAX to modulate apoptosis⁴², not like p107, which is shown to influence mitochondrial encoded gene expression. Moreover, p107 down regulation and genetic deletion is always associated with increased Oxphos^{2, 25, 26, 29, 43}, whereas Rb loss and reduced activity is attendant with both increased and decreased Oxphos that might be cell type or context dependent^{44, 45, 46, 47, 48, 49}.”

2.vii The strong effects observed on gene expression in Figure 2B for example would suggest that this might be a unique function of p107 (no obvious compensation by RB) but then would raise the question of the specific signals in the p107 protein that directs it to the mitochondria. Are the first 50-100 amino acids of p107 enough to form a mitochondria localization signal?

p107 does not possess a cleavable mitochondrial targeting amino acid pre-sequence (mts) located at the N-terminus as defined by the mitochondrial localization signal prediction software TargetP¹⁹. This was corroborated by direct visualization of Western blots that did not show a faster migrating cleaved form of p107 in the mitochondria. However, many mitochondrial localized proteins contain internal MTS like sequences rather than N-terminus cleavage sequences^{20,21}. Indeed, upon inspection, p107 possesses two potential internal mts-like sequences with high TargetP scores of 0.837 and 0.844 (a score >0.75 is very strong) (**new Suppl. Fig. 3**). We have added this data to the results section on page 5 as: “**Though p107 does not have an N-terminal mitochondrial targeting signal (mts), we found very strong scores (> 0.75) for putative internal mts-like sequences using the TargetP prediction algorithm³⁴, which normally predicts N-terminal pre-sequences (Suppl. Fig. 3)^{35, 36}.**” The new figures and experiments required us to update the Materials and Methods section.

Moreover, our findings for the presence of p107 in the mitochondria are centred on molecular, cellular and biochemical approaches that show it operating within this organelle as a transcriptional co-repressor. Biochemical mitochondrial fractionation showed that it is in the matrix where mtDNA resides and not on the outer or inner mitochondrial membranes. Moreover, confocal microscopy with subsequent Z-series analysis as well as Western blotting confirmed p107 localization within the mitochondria. qChIP analysis of mitochondrial lysates revealed that it interacted at the D-loop promoter region of mtDNA. Finally, p107 localization in the mitochondria is corroborated by a mitochondrial global proteomic study whose data base contains p107²².

2.viii Is the binding between SIRT1 and p107 dependent on p107 phosphorylation by Cdk2/4/6 (which could be tested by expressing a form of p107 that cannot be phosphorylated on the main Cdk2/4/6 targets, or possibly by pre-treating the protein extracts with a phosphatase before co-IP). It may be more difficult to determine whether SIRT1 deacetylates p107, but this could be another possibility.

We agree that this is a biologically important question. However, respectfully, we believe that assessing p107 post translational modification, which might affect its mitochondrial function is beyond the scope and main findings of this manuscript. To answer this question properly requires several data sets, experiments, and time to adequately confirm any finding. Furthermore, answering this question would open many new questions regarding post

translational modification and cell cycle, which will be the foundation of a complete study on its own. A PhD student to start in my lab (September 2021) will begin tackling this important question.

3.i The regulation of p107 mitochondrial activity by metabolic pathways is very exciting. Does the function of p107 in the nucleus (i.e. E2F binding) also change in response to changes in NAD⁺/NADH ratio and/or Sirt1 activity?

As per the reviewer's question we now show that the NAD⁺/NADH ratio is associated with the nuclear localization of p107. These data are now part of **revised Figure 3A**, which shows that a lower NAD⁺/NADH ratio (25mM glucose) compared to a higher NAD⁺/NADH (5.5mM glucose) is associated with decreased levels of p107 in the nucleus. For this result we have now added the following on page 9 of the Results section **“The presence of p107 in the mitochondria was inversely associated to the amount in the nucleus (Fig. 3A)**” We have also adjusted the Figure Legends appropriately.

3.ii Can the increased cell proliferation observed in p107 KO cells be rescued by decreasing ETC function downstream of p107 (e.g. through complex I inhibition with metformin)?

We thank the reviewer for this suggestion, as the new results stemming from his/her recommendation have strengthened our hypothesis. They further corroborate that p107 function in the mitochondria that affects cell cycle is linked to mitochondrial ATP generation. As requested, we have added metformin to p107KO cells to show inhibition of Oxphos potential (**new Figure 5M**) concomitant with reduced cell cycle and proliferation rate (**new Figure 5N and new Suppl. Fig. 27**). For these new results we now write on page 16 of the Results section: **“However, when Oxphos capacity was inhibited in p107KO cells with ETC complex 1 inhibitor metformin (Fig. 5M), the cell cycle and proliferation rate was significantly reduced (Fig. 5N and Suppl. Fig. 27).”** We have appropriately altered the figure legend and materials and methods for the new data set.

3.iii Figure 3A: how long were cells grown in the various concentrations of glucose and could the authors show total levels of p107? Are the differences due to changes in the cell cycle?

For Figure 3A, the cells were grown in stripped media containing 1mM, 5.5mM or 25mM for 20hrs. This information had been previously omitted erroneously and has now been added to the Supplemental Materials and Methods section. Also, as requested we now show total p107 levels in **revised Figure 3A** (the figure legends have been adjusted). We were unable to answer if the differences in p107 cellular localization were due to changes in cell cycle. This was due to an inability to G1-synchronize cells grown in stripped media with 5.5mM or 25mM glucose. This requires serum starvation for 3 days, which compromised the viability of the cells.

3.iv Loss or inactivation of Sirt1 seems to lead to an increase in p107 (Fig. 4B,E). Is this increase at the mRNA or protein level? Could this be a confounding factor that makes it appear as though p107 is translocating to the mitochondria, when in fact it is just more easily detected due to increased total expression?

As per the reviewer's request we have checked the p107 RNA expression pattern in Sirt1KO (**new Suppl. Fig. 15A**), nicotinamide treated c2MP cells (**new Suppl. Fig. 15C**) and c2MP and Sirt1KO cells grown in stripped media with 5.5 compared to 25mM glucose (**new Suppl. Fig. 15B & new Suppl. Fig. 10**). We found that the loss of Sirt1 (**Fig. 4B**) or the addition of Sirt1 inhibitor nicotinamide (**Fig. 4E**) or altering the glucose concentration (that is the NAD⁺/NADH ratio) (**Fig. 3A**) had no effect on the gene expression of p107. Thus, p107 transcription is not a confounding effect for p107 mitochondrial localization.

For these new data we have added to the end of the following sentence on page 12: "Unlike c2MPs grown in 5.5mM glucose that exhibit relocation of p107 from the mitochondria (**Fig. 3A**), Sirt1KO cells did not exhibit altered p107 mitochondrial localization (**Fig. 4B**)" the phrase "nor a change in p107 gene expression, including when grown in different glucose concentrations (**Suppl. Fig. 15A & 15B**)."
We have also altered the sentence on page 13 "We next determined if Sirt1 activity is necessary for p107 mitochondrial function. Inhibition of Sirt1 activity by nicotinamide (nam) increased p107 mitochondrial localization (**Fig. 4E**) that was concomitant with increased mtDNA promoter interaction (**Fig. 4F**)."
to read "We next determined if Sirt1 activity is necessary for p107 mitochondrial function. Inhibition of Sirt1 activity by nicotinamide (nam) increased p107 mitochondrial localization (**Fig. 4E**), which was concomitant with increased mtDNA promoter interaction (**Fig. 4F**). This occurred without a change in p107 gene expression (**Suppl. Fig. 15C**)."
Finally, for growth in different glucose conditions we added on page 9 "The presence of p107 in the mitochondria was inversely associated to the amount in the nucleus (**Fig. 3A**) and was not coupled to a change in p107 gene expression (**Suppl. Fig. 10**)."

3.v Figure 4M shows that resveratrol treatment increases mitochondrial gene expression even in the absence of p107, suggesting that either this drug has off-target effects, or that the link between Sirt1 activity and mitochondrial output occurs through additional factors other than p107. The authors should address this in the text.

It is well known that Sirt1 activity influences pro-oxidative mitochondrial output through additional factors such as activation of PGC-1 α ²³. In addition, these effects are compounded by our treatment of the cells with resveratrol over several hours, which would surely impact Sirt1 influence on additional pro-oxidative factors. To better ensure specific effects for Sirt1 on p107-specific mitochondrial function, we replaced this data set with new findings using the established Sirt1 activator srt1720 (**new Figures 4K, 4L, 4M, 4N, 4O & 4P**). The original resveratrol data (original Figures 4K, 4L, 4M, 4N and 4O) are now appended to the supplemental data section as new **Supplemental Figures 17A, 17B, 17C, 17D and 17E**.

For the new data sets we removed the following paragraph in the Results section: "c2MPs grown in a low concentration of res (10mM), had the opposite effect to nam for p107 localization and function. Treatment of c2MPs with this concentration of res decreased p107 within the mitochondria (**Fig. 4K**). The activation of Sirt1 also reduced p107 mtDNA promoter interaction and enhanced the mitochondrial gene expression (**Fig. 4L & 4M**), which corresponded to an increased ATP synthesis rate and capacity of isolated mitochondria (**Fig. 4N & Suppl. Table 1F**). No differences in ATP generation rate and capacity were observed in Sirt1KO c2MPs, which was anticipated with a non-significant consequence on mitochondrial gene expression (**Fig. 4M & 4O**). When Sirt1 activity was repressed by high doses of res, p107 was localized in the mitochondria and gene expression along with ATP generation rate and capacity were

significantly decreased (**Suppl. Fig. 7 & Suppl. Table 1G**).” In its place we have added the following paragraph on page 13 of the Results section:

“Next, we activated Sirt1 for a short window of time. Treatment of c2MPs with Sirt1 activator, srt1760, for 3 hours resulted in decreased p107 protein levels within the mitochondria (**Fig. 4K**). The activation of Sirt1 also reduced p107 mtDNA promoter interaction (**Fig. 4L**) and enhanced the mitochondrial gene expression, but not in p107KO and Sirt1KO cells (**Fig. 4M**). The increase in gene expression with Sirt1 activation within this short time window was associated with increased ATP generation (**Fig. 4N**). Importantly, relative mtDNA copy number was not affected by this short-term treatment (**Fig. 4O**), suggesting that mitochondrial gene expression and not mtDNA copy number contributed to the differences in ATP generation. ATP generation rate and capacity were not altered by srt1760 treatment in Sirt1KO c2MPs (**Fig. 4P**). We also confirmed the importance of Sirt1 activity to p107 mitochondrial function by using resveratrol at low (**Suppl. Fig. 17 & Suppl. Table 1F**) and high doses (**Suppl. Fig. 18 & Suppl. Table 1G**) that indirectly activate and inhibit Sirt1 activity, respectively.”

The new figures and experiments required us to update the Figure Legends and Materials and Methods sections.

3.vi The Seahorse assays in Fig. 5F and Supp. Fig. 13B show that the presence or absence of p107 alters ATP production from both glycolysis and the mitochondria. Why is glycolytic output changing, when the author’s model places p107 downstream of glycolysis? Is this an indirect effect, or is p107 influencing metabolism through multiple mechanisms? Because ATP output from both pathways is affected by changes in p107 levels, one cannot conclude that it is specifically p107’s regulation of oxphos that is influencing cell proliferation.

The reviewer has brought up a great question that we now address in the discussion in the manuscript. Recent findings show that glycolytic output is tied to Oxphos output²⁴. This is linked to the availability of NAD⁺, such that increasing Oxphos capacity increases glycolytic output by increasing NAD⁺ levels, and vice versa. We also show the same phenomenon. These experiments were performed by increasing pyruvate oxidation that increases NADH levels by the TCA. The increased levels of NADH in this situation cannot be oxidized sufficiently by the ETC, thereby reducing NAD⁺ availability^{24,25}. Our use of oxamate also decreased the NAD⁺/NADH ratio (**Fig. 3J**) thus increasing NADH availability, concomitant with a reduced cell cycle potential (**Fig. 5J, Fig. 5K & Fig. 5L**) and Oxphos capacity (**Fig. 5I**). Thus, the rationale for an increase in glycolytic activity in p107KO is greater NAD⁺ availability, as a consequence of more effective oxidation of NADH by Oxphos. Thus, ATP generation increases from both glycolysis and Oxphos. Oppositely, with the addition of p107fl or p107mls that down regulates Oxphos capacity, the availability of NAD⁺ is reduced, which reduces the glycolytic output (**Suppl. Fig. 25**). Hence, ATP generation decreases from both glycolysis and Oxphos. When we blocked the Oxphos capacity with metformin in p107KO cells, we reduced the ATP generated from Oxphos and ultimately glycolysis (**new Fig. 5M**), resulting in a reduction in cell cycle (**new Fig. 5N and Suppl. Fig. 27**).

To highlight how p107 might affect glycolysis, the following changes to the manuscript have been made:

a) We removed the following from the Introduction section “In glycolysis, ATP is produced at a fast rate and the glycolytic pathway components might be used for the biosynthesis of nucleic acids, proteins, carbohydrates and lipids essential for cell proliferation²⁶. On the other hand,

Oxphos, which produces at least 10 times more ATP, is crucial for progression through the G1/S transition of the cell cycle²⁷. Hyperactivation of Oxphos in proliferating cells is critical for their advancement, whereas its down regulation delays or blocks progression to S phase²⁸⁻³⁰.” And “NADH is a by-product of glycolysis that might be used as a reducing reagent required for Oxphos whereas, NAD⁺, a coenzyme in various metabolic pathways, is the oxidized form of NADH that can be made in glycolysis from the transformation of pyruvate to lactate” and added the following on page 2: “NADH is a by-product of glycolysis that can be indirectly transferred to the mitochondria via the Malate-Aspartate shuttle to be used in Oxphos as reducing equivalents. Alternatively, NADH might be oxidized to NAD⁺ in the reaction that produces lactate from pyruvate, the end product of glycolysis.

NAD⁺ is an important coenzyme in several metabolic pathways and has been demonstrated to be essential for proliferation^{7, 8, 9, 10, 11},”

b) We also added the following paragraph to the Discussion section on page 20 to highlight that increase in Oxphos capacity affects cell cycle rate by influencing the glycolytic rate.

“Our results demonstrate that increased Oxphos is linked to an increase in cell cycle rate in p107KO compared to control cells that might be due to an enhanced supply of free NAD⁺^{7, 8}. Recently, it was shown that promoting pyruvate oxidation by PDK inhibition reduced the NAD⁺/NADH ratio concomitantly with decreased cellular proliferation and ATP generation⁷. We also show increasing pyruvate oxidation with oxamate, DCA or Ldha KD, lowered the NAD⁺/NADH ratio, Oxphos potential and cell cycle rate. Importantly, we found that these effects were tied to p107 function in the mitochondria that decreased mitochondrial gene expression. It is likely that NADH is more effectively oxidized in the p107KO cells, increasing ATP generation from Oxphos that increases NAD⁺ accessibility promoting a faster cell cycle. Additionally, for p107KO cells, glycolysis produces more ATP with increased NAD⁺ availability. Furthermore, p107KO cells are not affected by NADH overload to the mitochondria, with increased pyruvate oxidation by addition of oxamate, DCA or Ldha KD. We propose this is due to a greater capacity for NADH oxidation by Oxphos in p107KO cells compared to controls. Indeed, blocking the ETC cycle in p107KO cells with metformin reduced the cell cycle rate, thus emphasizing the importance of Oxphos potential to cell cycle.”

Reviewer #2 (Remarks to the Author):

In this manuscript, the authors investigate the role of p107 in mitochondrial metabolism. p107 is a repressor of E2F transcription factors in the mitotic cell cycle. There are very few functions that are known to be specific to p107 compared to RB and p130. The authors have carefully investigated the localization of p107 in mitochondria, its binding to (mitochondrial) chromatin, gene expression, and metabolism. The interaction of p107 with SIRT1 is interesting and novel.

The authors present a lot of data and in most cases appropriate controls are included. Overall, it is an interesting story but to some degree it is very p107-centric. The authors should tone down their claims since it is very well known that p107 functions in repressing E2F (which is known to affect metabolism – see L. Faja’s work). I would encourage the authors to be more cautious with the interpretation of their data and how to put it into context with what has been published before.

These are the issues, which need to be addressed before this manuscript can be considered:

1. In the introduction the authors indicate that proliferating cells need more ATP and non-

proliferating cells. For a long time, I was thinking the same until I discussed this with Matthew Vander Heiden. He had made measurements and found that this was not true. Therefore, I would recommend that the authors use caution in that respect.

We very much thank the reviewer for bringing this to our attention, we meant to say that ATP was necessary for cell division and not cell cycle rate. Thus, we removed the word “rate” from the sentence on page 2 “Indeed, the cell cycle rate is dependent on the amount of total ATP generated from glycolysis and Oxphos” to read “**Indeed, the cell cycle is dependent on the amount of total ATP generated from glycolysis and Oxphos**”

Also to further reduce the confusion regarding the association between increasing Oxphos and cell cycle rate, we have removed the following in the Introduction section “In glycolysis, ATP is produced at a fast rate and the glycolytic pathway components might be used for the biosynthesis of nucleic acids, proteins, carbohydrates and lipids essential for cell proliferation²⁶. On the other hand, Oxphos, which produces at least 10 times more ATP, is crucial for progression through the G1/S transition of the cell cycle²⁷. Hyperactivation of Oxphos in proliferating cells is critical for their advancement, whereas its down regulation delays or blocks progression to S phase²⁸⁻³⁰.”

Furthermore, Dr. Vander Heiden’s recent findings this year^{24,25} showcase that proliferating cells actively reduce Oxphos under conditions of higher glycolysis. We have thus added this reference to the statement on page 2 “**Proliferating cells customize methods to actively reduce Oxphos under conditions of higher glycolysis**^{6,7}.”

We also found that decreasing the NAD⁺/NADH ratio reduced the rate of cell cycle, as did Dr. Vander Heiden’s group²⁴. To more to better highlight the importance of the NAD⁺/NADH ratio we removed the statement: “Whereas, NAD⁺, a coenzyme in various metabolic pathways, is the oxidized form of NADH that can be made in glycolysis from the transformation of pyruvate to lactate,” and added the following to the introduction section on page 2 with appropriate references: **NAD⁺ is an important coenzyme in several metabolic pathways and has been demonstrated to be essential for proliferation**^{7, 8, 9, 10, 11}.

Finally, we also added the following paragraph in the Discussion section on pages 20-21 to highlight that increase in Oxphos capacity and cell cycle is related to the NAD⁺ availability as per Dr. Vander Heiden’s recent findings.

“Our results demonstrate that increased Oxphos is linked to an increase in cell cycle rate in p107KO compared to control cells that might be due to an enhanced supply of free NAD⁺ ^{7, 8}. Recently, it was shown that promoting pyruvate oxidation by PDK inhibition reduced the NAD⁺/NADH ratio concomitantly with decreased cellular proliferation and ATP generation⁷. We also show increasing pyruvate oxidation with oxamate, DCA or Ldha KD, lowered the NAD⁺/NADH ratio, Oxphos potential and cell cycle rate. Importantly, we found that these effects were tied to p107 function in the mitochondria that decreased mitochondrial gene expression. It is likely that NADH is more effectively oxidized in the p107KO cells, increasing ATP generation from Oxphos that increases NAD⁺ accessibility promoting a faster cell cycle. Additionally, for p107KO cells, glycolysis produces more ATP with increased NAD⁺ availability. Furthermore, p107KO cells are not affected by NADH overload to the mitochondria, with increased pyruvate oxidation by addition of oxamate, DCA or Ldha KD. We propose this is due to a greater capacity for NADH oxidation by Oxphos in p107KO cells compared to controls. Indeed, blocking the ETC cycle in p107KO cells with metformin reduced the cell cycle rate, thus emphasizing the importance of Oxphos potential to cell cycle.”

2. Figure 1: it seems that only a small fraction of p107 is localized to the mitochondria judging from WB (1B). In the confocal analysis, almost all p107 is localized to the mitochondrial and the signal in the nucleus is almost undetectable (1D). Somehow these results do not add up. Can the authors comment on this? BTW, why did the authors not use mitotracker as a marker for mitochondria?

We believe that the reason for the p107 level discrepancies in the mitochondria between the Western blotting and confocal microscopic analysis has to do with differences in the approaches used. First, we used different antibodies for microscopic analysis (monoclonal anti-p107 SD9, Santa Cruz Biotech) versus Western blotting (polyclonal anti-p107 13354-1-AP, Proteintech). Thus, the availability of these epitopes under the two experimental manipulations may have resulted in different detection sensitivities. For example, the fixing protocol used in confocal microscopy may not have been optimal for detection of nuclear p107.

Second, and most importantly, the yield of protein from mitochondrial lysates is very low due to our stringent isolation procedure. This is due to maximising the purity of the mitochondrial fractions (which are mitoplasts) for Western blotting with several washes to eliminate any contaminating cytoplasmic proteins in this fraction. In fact, p107 protein levels are very high outside of the nucleus. We found by normalizing total p107 protein levels on Western blots that there is as much as 73% outside the nucleus in asynchronous proliferating cells (**new Suppl. Fig. 2**). For these data we write on page 5: “Of the total p107 protein that was expressed in proliferating c2MPs, about 28% and 45% were present in the mitochondria and cytoplasmic compartments, respectively (Suppl. Fig. 2).” The Material and Methods have been updated for this new information. Finally, there is a lack of p107 in the nucleus during G1¹⁻⁵ that we have substantiated with our **new Figures 1M & Fig. 1N**.

We initially did not use MitoTracker to show co-localization because we had experimental issues with bleeding of the dye into the nucleus. This technical shortcoming has now been surmounted in this revised manuscript and represented by the **new Supplemental Figure 4**. We now write on page 6:

“The protein localization results were corroborated by confocal microscopy and subsequent analysis of generated z-stacks that showed p107 and the mitochondrial protein Cox4 and MitoTracker Red co-localize in c2MPs (**Fig. 1E & Suppl. Fig. 4A**). The specificity of immunocytochemistry was confirmed by immunofluorescence of p107 in p107 “knockout” c2MPs (p107KO c2MPs) generated by Crispr/Cas9 (**Suppl. Fig. 5A**). Moreover, p107 and Cox4 or MitoTracker Red fluorescence intensity peaks were matched on a line scanned RGB profile (**Fig. 1F & Suppl. Fig. 4B**) and orthogonal projection showed co-localization in the XY, XZ, and YZ planes (**Fig 1G & Suppl. Fig. 4C**). The same assays were used to confirm these findings in proliferating primary wild type (Wt) MPs (prMPs) and p107 genetically deleted prMPs (p107KO prMPs) isolated from Wt and p107KO mouse skeletal muscles, respectively (**Fig. 1H, 1I, 1J & Suppl. Fig. 4D, 4E & 4F**). In vivo, in a model of regenerating skeletal muscle caused by cardiotoxin injury, we also found p107 co-localized with Cox4 in proliferating MPs, which express the myogenic stem cell marker Pax7 (**Fig. 1K, 1L & Suppl. Fig. 6**). Together, these data showcase that p107 localizes in the mitochondria of MPs from primary and muscle cell lines, suggesting that it might have an important mitochondrial function in the actively dividing cells.” For this new data we have also updated the Materials and Methods section.

3. Figure 2A: in the ChIP experiments, the y-axis is “Promoter Occupancy” which is unusual. Most scientist will use percent compared to input. Do they authors have a specific reason why they are not following convention?

We have changed the y-axis to “Fold Enrichment” from “Promoter Occupancy” for the ChIP data figures in the manuscript, which is a better interpretation of the results obtained. Our data describe the interaction at the promoter by normalizing the relative occupancy to IgG (negative base line control). This was determined by amplifying isolated DNA fragments by qPCR from the chromatin immunoprecipitation of mitochondrial lysates for p107, Tfam, E2f4, and IgG using the D-loop primer sets (**Suppl. Table 2**) to obtain Ct values. Using these values, the fold changes were determined for p107, Tfam, and E2f4 normalized to IgG Ct. We have also changed the wording in the Materials and Methods and Supplemental Materials and Methods section to better reflect how we arrived at the data for the ChIP experiments.

4. Figure 2B-D: these results are interesting but do not prove that p107 is responsible for these changes. It is correlative. The authors should be cautious when they describe and interpret these kinds of data.

As per the reviewer’s suggestion to be more in tune with a correlative finding that mtDNA gene expression changes with p107 mitochondrial levels, we changed the wording of a subheading and subsequent lead sentence to its subsection.

The subheading on page 7 is changed from “p107 represses mtDNA encoded gene expression to regulate Oxphos capacity” to “**p107 is associated with lowered mtDNA encoded gene expression**” The lead sentence to the sub-section has been changed from “We next appraised if p107 mtDNA promoter occupancy might repress mitochondrial encoded gene expression.” to “**We next appraised if p107 mtDNA promoter occupancy might influence mitochondrial encoded gene expression.**”

We added the next subheading on page 8 to read “**p107 mitochondrial function is associated with Oxphos capacity**”

5. Figure 3A-B: is it glucose dependent or proliferation dependent? This is important since higher glucose levels could stimulate proliferation.

Higher glucose levels do stimulate proliferation but not in stripped media (ie. media lacking pyruvate and glutamine) where S-phase is significantly reduced (**new Fig. 5F and new Suppl. Fig. 24**). Please read the answer to the next comment.

6. Figure 3F: this result is counterintuitive unless high glucose levels are toxic for these cells. The authors should explain this more carefully.

Yes, we agree with the reviewer that this finding appears to be in contradiction with what the literature has reported regarding the role of glucose concentration and cell cycle rate. We were initially taken aback by the findings. However, the difference between the findings in the literature and our own is that we performed the experiment in stripped media with glucose as the sole nutrient. In these cells, the shift to more NADH with higher glucose reduces the availability

of NAD⁺ from Oxphos (we postulate this is due to p107) that is required for cell cycle²⁴. We found S-phase of the cell cycle is significantly reduced in 25mM compared to 5mM glucose, without glutamine and pyruvate (**new Fig. 5F**). This contrasts with an increase in cell cycle rate when pyruvate and glutamine are added to the cells grown in 25mM compared to 5mM glucose (**new Suppl. Fig. 24**), which is in line with what has been reported in the literature. To the Results section on page 15 we added our new data that S-phase of the cell cycle is significantly reduced in 25mM compared to 5mM glucose, without glutamine and pyruvate: “**Second, we tested when endogenous p107 is significantly expressed in the mitochondria by growing cells in stripped media containing 25mM compared to 5.5mM glucose (Fig. 3A). In concordance with p107 mitochondrial localization, we found that S-phase was significantly reduced for cells grown only in 25mM glucose compared to 5.5mM (Fig. 5F and Suppl. Fig. 24A). Contrarily, S-phase was significantly increased for cells grown in complete media containing 25mM glucose compared to 5.5mM or cells grown (Suppl. Fig. 24B). These findings suggest that p107 might direct the cell cycle rate through management of Oxphos ATP generation**” For these data Figure Legends and Materials and methods section has been altered.

7. The relation of p107 and the NAD⁺/NADH ration [probably better to calculate the NAD⁺/(NADH+NAD⁺) ratio] is a chicken and egg question. I would suggest that the authors are careful with this.

We believe that the mitochondrial function of p107 is downstream of the influence of NAD⁺/NADH from two main avenues of investigation. First, we have showed by several experiments that p107 is influenced by the NAD⁺/NADH ratio and not vice versa. In Figure 3 we provide evidence that this modulation of this ratio (whether through galactose or differing glucose levels or treatment with oxamate, DCA or Ldha knockdown) influences p107 localization (**new Suppl. Fig 12**). Moreover, when glutamine is altered, the NAD⁺/NADH ratio is not altered nor is there a difference in p107 cellular compartmentalization. Second, experiments in Figure 4 show convincingly that p107 function is downstream of Sirt1 activity that is controlled by NAD⁺. Inhibiting Sirt1 with nicotinamide increased p107 mitochondrial localization and activating Sirt1 with srt1720 (**new Figures 4K, 4L, 4M, 4N, 4O & 4P**) decreased the levels of p107 in the mitochondria. We do recognize that the changes in Oxphos that are coordinated by p107 will ultimately influence NAD⁺/NADH.

Reviewer #3 (Remarks to the Author):

The authors are motivated by their interest in the metabolic states which govern muscle stem cells proliferation and differentiation. Thus, they focus on the regulation of glycolysis versus OXPHOS. In the frame of this interest they identified p107 – a member of the RB protein family, which is here shown to localized in the mitochondria of mouse cells. Previously (2017), the same authors showed association between reduced expression of p107 and increase in OXPHOS protein levels and function in human muscle in response to exercise. This suggested a repressive activity of p107 on mitochondrial function. In the current manuscript, the results indeed demonstrate that mitochondrial localization of p107 associate with reduced mtDNA gene expression in several primary and transformed mouse cells. Therefore, one can view this as a follow-up study. Secondly, the authors suggest that

the involvement of p107 in mtDNA gene expression occurs via mtDNA binding at the D-loop based on ChIP-qPCR results. They show that such reduced gene expression increases the length of the cell doubling time, and reduce mitochondrial ATP production. Then, they show evidence that the very mitochondrial localization of p107 negatively correlates with the expression of SIRT1, i.e. SIRT1 expression attenuates mitochondrial localization of p107 which leads to expression of mtDNA genes (with a continuous mitochondrial localization of p107 upon SIRT1 KO).

In general, the results in this manuscript are impressive, encompassing gene manipulation in cells (primary cell culture and immortalized cells) and in the whole organism (mouse). In my personal opinion, showing that an RB-interacting protein (p107) directly affects mitochondrial activity is very important, as it offers connection of cell cycle regulation to mitochondrial regulation. Nevertheless, there are several points that I feel should be revised to improve the manuscript:

1. Although much information is given regarding the impact of reduced mtDNA gene expression in response to mitochondrial localization of p107, I missed discussion and experiments which relate to the mechanism by which such reduced expression occurs. In other words, for the past five decades much data accumulated regarding the core elements and factors that govern mtDNA transcription. These factors are not mentioned at all in the current manuscript, namely the RNA polymerase of the mitochondria - POLRMT, transcription factor A – TFAM, TFB2M, mitochondrial transcription elongation factor – TEFM and mitochondrial transcription termination factors MTERF. There are multiple papers published in this field including relatively recent reviews in good journals. Here are two major examples: Gustaffson et al Annual Rev of Biochemistry (2016); Barshad G et al Trends in Genetics, 2018.

How is the impact of p107 on mtDNA gene expression relates to the mentioned factors and core mechanism of mtDNA transcription? It is essential to discuss p107 in the context of mtDNA transcriptional regulation and its relations with such regulatory factors.

We thank you very much for your positive comments, we really appreciated your feedback and have done our best to answer your concerns.

We had originally introduced and discussed the core elements of mitochondrial gene transcription in the Introduction and Discussion sections, but these were eliminated in our final draft to reduce the number of characters in the manuscript. As per the reviewer's request, we have now added back to the Introduction and Discussion, aspects for how the mitochondrial transcriptional machinery is associated with p107 mitochondrial function. In the Introduction section on page 3 we now write:

“It is uncertain if the mtDNA transcription initiation machinery comprises transcription factor B2 of mitochondria (Tfb2m) and mitochondrial DNA-directed RNA polymerase (Polrmt) or a three component system that also includes transcription factor A of mitochondria (Tfam)^{20, 21, 22}. Besides the initiation factors, the transcription machinery also includes the mitochondrial transcription elongation factor (Tefm), that ensures processivity and stabilization of the elongation complex and termination of transcription performed by mitochondrial transcription termination factor 1 (Mterf1)¹⁷. Multiple reports have suggested that transcription factors and co-transcriptional regulators that are typically functional in the nucleus also have regulatory functions in the mitochondria²³. However, relative to nuclear gene expression, regulation of mitochondrial gene expression is poorly understood.”

In the Discussion section on pages 19 we now write: “Regulation of mitochondrial gene expression is poorly understood relative to control of nuclear gene transcription. The involvement of known regulators of gene expression in the nucleus to play a role in governing mtDNA gene expression via actual localization in the mitochondria has been shown for several factors²³. It is currently not definitive if p107 influences mitochondria encoded gene expression and/or mitochondrial DNA copy number by interacting within the regulatory D-loop region or outside at other region(s) along the mtDNA, as has been shown for other regulators^{23, 50, 51}. Moreover, it is not clear if the mitochondrial transcriptional regulators, including p107, operate in a cell or physiological specific context. However, our findings link a nuclear co-transcriptional repressor directly to the repression of mitochondrial transcription, which influences metabolism and cell cycle.”

More importantly, as mandated by the reviewer, we have now detailed how p107 might control mtDNA gene expression as it relates to nuclear and mitochondrial transcription factors. Our new results show that p107 mitochondrial function is associated with its nuclear transcription factor binding partner E2f4 (**new Fig. 2B and new Fig. 2C**). By Western blot, we found that E2f4 but not E2f5 is present in the mitochondria in proliferating cells (**new Fig. 2B**). Moreover, by ChIP assays E2f4 enrichment at the mtDNA is evident in proliferating cells but barely in growth arrested cells (**new Fig. 2C**). Interestingly, in proliferating c2MPs, our results show that a very small amount of Tfam is present in the mitochondria (**new Fig. 2B**) which does not interact at the mtDNA (**new Fig. 2C**). This contrasts with growth arrested cultures (**new Fig. 2B and new Fig. 2C**). Note that mouse specific ChIP-validated POLRMT and TFB2M antibodies are not presently commercially available nor has their use been published. For these new data on page 7 of the Results section we write:

“As p107 does not directly interact with DNA, we assessed potential interacting transcription factors in the mitochondria. We evaluated potential mitochondrial role for the putative mitochondrial transcription factor Tfam, as well as the p107 interacting nuclear transcription factors E2f4 and E2f5³¹. By Western blotting, we found that E2f4 was present in proliferating mitochondrial lysates, compared to negligible levels of Tfam and the complete absence of E2f5 (**Fig. 2B**). Moreover, ChIP analysis indicated that E2f4, but not Tfam, interacted at the mtDNA of proliferating c2MPs, whereas the opposite pattern of interaction was found during growth arrest (**Fig. 2C**). Intriguingly, these results suggest that E2f4 might be a mitochondrial binding partner of p107 at the mtDNA during proliferation of c2MPs.” These new data required us to update Figure legends and the Materials and Methods section.

2. In addition, the involvement of a known regulator of gene expression in the nucleus also in the regulation of mtDNA gene expression via actual localization in the mitochondria has been shown for several factors, such as MEF2D (She et al. JCI 2011), MOF (Chatterjee et al. Cell 2016) and NFATC (Lambertini et al The International Journal of Biochemistry & Cell Biology 64 (2015) 212–219). The claim that p107 affects mtDNA gene expression via mitochondrial localization and possibly via mtDNA binding should be argued in the context of such factors, and others (all summarized in the review paper that I already mentioned above - Barshad G et al Trends in Genetics, 2018). All these papers including this review, should be cited in the right context (introduction and discussion sections).

As requested by the reviewer we have now added these papers in the context of the Discussion section on pages 19: “Regulation of mitochondrial gene expression is poorly understood relative

to control of nuclear gene transcription. The involvement of known regulators of gene expression in the nucleus to play a role in governing mtDNA gene expression via actual localization in the mitochondria has been shown for several factors²³. It is currently not definitive if p107 influences mitochondria encoded gene expression and/or mitochondrial DNA copy number by interacting within the regulatory D-loop region or outside at other region(s) along the mtDNA, as has been shown for other regulators^{23, 50, 51}. Moreover, it is not clear if the mitochondrial transcriptional regulators, including p107, operate in a cell or physiological specific context. However, our findings link a nuclear co-transcriptional repressor directly to the repression of mitochondrial transcription, which influences metabolism and cell cycle.”

3. Inaccuracies: in the 2nd paragraph of the introduction the authors claim the following: "NADH is a by-product of glycolysis that might be used as a reducing reagent required for Oxphos". This point is incorrect: NADH for the OXPHOS is generated in the frame of the TCA cycle by Malate dehydrogenase. NADH from glycolysis cannot be imported into the mitochondria as there is no import machinery for NADH, yet its electrons can be indirectly imported from the cytosol to the mitochondria. The authors are asked to revise the sentence while taking into account the mentioned facts.

Thanks for bringing this to our attention. We meant to say that the electrons from NADH can be indirectly imported from the cytosol to the mitochondria via Malate dehydrogenase.

We have changed the incorrect sentence on page 2 from "NADH is a by-product of glycolysis that might be used as a reducing reagent required for Oxphos" to " **NADH is a by-product of glycolysis that can be indirectly transferred to the mitochondria via the Malate-Aspartate shuttle to be used in Oxphos as reducing equivalents.**"

4. Throughout the manuscript the authors consider the impact of SIRT1 on p107, while not considering at all the other 6 members of the SIRTUIN family, especially SIRT3 which has a known mitochondrial localization and activity. The authors should relate to the entire SIRT family and justify why SIRT1 is the only one mentioned and manipulated.

As recommended by the reviewer, we introduced the Sirt family in the Introduction section on page 2 by adding the following:

“**NAD⁺ is an important coenzyme in several metabolic pathways and has been demonstrated to be essential for proliferation^{7, 8, 9, 10, 11}. Importantly, it is an activator of seven sirtuin protein family members (Sirt1-7), which are deacetylases that target several proteins¹². The sirtuins operate in specific cellular locations with non-redundant substrate preferences. Sirt3, Sirt4, and Sirt5 are located in the mitochondria, Sirt6 and Sirt7 in the nucleus and Sirt1 and Sirt2 are found in both the nucleus and cytosol¹³. Of these, Sirt1 is the most well-studied family member operating as an energy sensor of the cytoplasmic NAD⁺/NADH ratio¹². It regulates metabolic homeostasis by enhancing mitochondrial metabolism through activation by cytoplasmic NAD⁺¹¹.**”

Also, as requested by the reviewer, we justified why we chose to study Sirt1-p107 interaction over the other Sirt family members by changing the sentence in the Results section on page 12 : “As the NAD⁺ dependent Sirt1 deacetylase is an energy sensor of the NAD⁺/NADH ratio, we

evaluated if it potentially regulates p107.” to “As the NAD⁺ dependent Sirt1 deacetylase is an energy sensor of the NAD⁺/NADH ratio, found in the cytoplasm and targets several transcription factors and co-activators¹², we evaluated if it potentially regulates p107”.

5. Binding experiments of p107 to the mitochondrial genome were performed using ChIP-qPCR, focusing on the D-loop. This experiment assumes that protein binding there will affect the regulation of the mitochondrial genome. However, factors that bind the mtDNA in addition to the core regulators of mtDNA transcription (as mentioned in comment 2), may bind outside of the D-loop, yet affect mtDNA transcription (see especially the papers about MEF2D and MOF – mentioned above in comment 2). In addition, screen of ChIP-seq experiments available from ENCODE revealed mtDNA binding by additional known regulators of nuclear gene expression, which were experimentally shown to localize both in the nucleus and in the mitochondria in human cells (Blumberg et al. 2014 Genome Biol and Evol). The experiment done by the authors assumes D-loop binding by p107, which is not necessarily the case. Hence, the impact of p107 on mtDNA gene expression does not necessarily occur via D-loop binding – we will not know that until ChIP-seq experiments are performed. This point should be mentioned by the authors in the manuscript.

As requested by the reviewer we now state that it is not known for sure where p107 interacts along the mtDNA and we have cited other findings where regulators have been shown to interact at different locations on the mtDNA, as per his/her suggestion. We now write on page 19 in the Discussion section “It is currently not definitive if p107 influences mitochondria encoded gene expression and/or mitochondrial DNA copy number by interacting within the regulatory D-loop region or outside at other region(s) along the mtDNA, as has been shown for other regulators^{23, 50, 51}.”

6. The impact of p107 on mitochondrial gene expression does not consider coordination between mtDNA and nuclear DNA-encoded OXPHOS genes, which could be easily measured by RNA-seq experiments of at least some of the tested cells. This should be performed.

This is a good suggestion and easy to perform, but unfortunately, we currently do not have sufficient funds to run these experiments. Nonetheless, detailing nuclear DNA-encoded Oxphos genes is somewhat beyond the scope of this paper and would raise many follow-up questions that we think would be best addressed in a full separate study. One of these important questions is how retrograde signalling from mitochondria to the nucleus is being controlled by p107 mitochondrial function. Also, the additional amount of data would result in less emphasis on our key finding for p107 function in the mitochondrial encoded gene expression.

7. In the bottom of page 5 the authors wrote: "The importance of p107 to mitochondrial gene expression was confirmed with p107KO c2MPs and prMPs, which both exhibited significantly increased mitochondrial encoded gene expression in the genetically deleted cells compared to their controls (Fig. 2C)." Increased expression of mtDNA-encoded genes is indeed shown, but it could be due to overall altered numbers of mitochondria in the cell, not necessarily due to regulation of transcription. This should be assessed by measurement of mtDNA copy number. With this in mind, in page 8 the authors

wrote: "Moreover, the mtDNA to nuclear DNA ratio remained unchanged when grown in the different glucose concentrations, in contrast to MPs treated with the mitochondrial biogenesis activator AICAR (Fig. 3E)." This is totally confusing: mtDNA to nuclear DNA ratio measurement seems to me assessment of mtDNA copy number; why was it not assayed in the p107 KO? The authors are asked to add this assay here as well and interpret the impact of p107 appropriately.

Thanks for bringing this to our attention. We understand the reviewer's confusion concerning mtDNA copy numbers, as this was not written properly. To eliminate any confusion, we have now replaced the phrase on pages 8, 10 and 13 from "...the mtDNA to nuclear DNA ratio..." with "**the relative mtDNA copy number...**". We have also changed the y axis on the graphs with mtDNA copy number data from "mtDNA/nDNA" to "**relative mtDNA copy #**" to better reflect the nature of the findings.

We had originally not assayed the p107KO cells because we believed showing the effect in control cells was sufficient. However, as requested by the reviewer we have now performed the requested experiment. We found that p107KO cells had significantly more mtDNA than controls (**new Fig. 2H**). For these new data we write on page 8: "**The increased ETC complex formation might not only signify differences in mtDNA transcription, as we also found that relative mtDNA copy number was significantly greater in p107KO cells compared to controls (Fig. 2H).**"

However, the differences in ATP generation rate and capacity between control and p107KO are reflective of differences in ETC levels per mitochondria rather than mitochondria number, as ATP generation measurements were normalized to mitochondrial protein content (**Fig. 2J**). To emphasize this on page 8 we write: "**To exclude the potential contribution of mitochondrial biogenesis, measurements were normalized to mitochondrial protein content.**" To put this in perspective with the ATP generation rate increase in p107KO cells we write on page 8: "**Although mtDNA copy number, which typically is associated with mitochondria content, was significantly greater in p107KO cells (Fig. 2H), the differences in ATP generation rate and capacity between control and p107KO cells are reflective of differences in ETC levels per mitochondria rather than mitochondria number.**" And on page 9: changed the sentence from "Together these data suggest that p107 has repressor activity when bound to the mtDNA promoter reducing the capacity to produce ETC complex subunits, which influences the mitochondria potential for ATP generation." to "**Together these data suggest that p107 has repressor activity when interacting at the mtDNA promoter regulating mtDNA copy number and/or gene expression to reducing the capacity to produce ETC complex subunits, which influences the mitochondria potential for ATP generation.**" For these data we have updated the Figure legends and Materials and Methods sections.

Moreover, the levels of mtDNA copy number did not change for the p107KO cells in different glucose concentrations (**new Fig. 3F**). For these new data, on page 9 we removed "Together, these data show that mitochondrial gene expression might be mediated by glycolytic flux induced p107 mitochondrial gene repression and not merely a consequence of differences in mitochondrial biogenesis." and added "**We also found that the relative levels of mtDNA did not change in p107KO cells when grown in the different glucose concentrations (Fig. 3F). Together, these data show that the regulation of mitochondrial encoded gene expression by glucose**

concentrations is not due to altered mtDNA number and/or mitochondrial biogenesis within the time period of the experiments.” Figure Legends and Materials and Methods section has been updated for these new findings.

Finally, when we treated cells for a short period of time (3hrs) with Sirt1 activator srt1720 (**new Fig. 4**) we found that mitochondria encoded gene expression leading to influence on Oxphos capacity was dependent on p107. In this small window of time, mtDNA replication is unlikely.

8. In page 7 of the manuscript the authors wrote: "Interestingly, we found that p107KO c2MPs had an elongated mitochondrial network (Fig. 2I) made up of mitochondria with significantly increased length and area (Fig. 2J)." Aside from mitochondrial fusion, such elongation could also be due to increased numbers of mitochondria (see previous comment) which under the microscope may appear as elongated structures. The authors are asked to refer to such possibility.

As requested by the reviewer on page 9 we now write: “**Aside from mitochondrial fusion, the elongation could also be due to increased numbers of mitochondria that may appear as elongated structures under the microscope.**”

Minor comments

1. In the 2nd paragraph of the introduction the authors wrote: "Whereas, NAD⁺, a coenzyme in various metabolic pathways, is the oxidized form of NADH that can be made in glycolysis from the transformation of pyruvate to lactate". This sentence has no end – please re-write; also it has to be corrected with respect to the previous sentence about NADH: NAD⁺ is a cofactor of the SIRTUINS inside the mitochondria (SIRT3) - using the NAD⁺ from the TCA, and NAD⁺ from the glycolysis is used as cofactor for the rest of the Sirtuins outside of the mitochondria.

As requested by the reviewer we have removed the sentence “Whereas, NAD⁺, a coenzyme in various metabolic pathways, is the oxidized form of NADH that can be made in glycolysis from the transformation of pyruvate to lactate.” and added on page 2 “**NADH is a by-product of glycolysis that can be indirectly transferred to the mitochondria via the Malate-Aspartate shuttle to be used in Oxphos as reducing equivalents. Alternatively, NADH might be oxidized to NAD⁺ in the reaction that produces lactate from pyruvate, the end product of glycolysis.**”

We have also altered the sentence on page 2: “High levels of cytoplasmic NAD⁺ activate the lysine sirtuin (Sirt) deacetylase family, including Sirt1” to “**NAD⁺ is an important coenzyme in several metabolic pathways and has been demonstrated to be essential for proliferation^{7, 8, 9, 10, 11}. Importantly, it is an activator of seven sirtuin protein family members (Sirt1-7)....**”

2. In page 16 the authors wrote: "Thus, the decreased proliferative capacity of p107 over expressing cells found in many reports might now be attributed to a mitochondrial role in repressing mitochondrial gene expression, which is required for G1 cell cycle progression." This statement may reflect over-interpretation of the data, as p107 could also be found in the nucleus with unknown function there.

As requested by the reviewer we changed the indicated sentence to reduce the over interpretation of the data on page 20 from: “Thus, the decreased proliferative capacity of p107 over expressing cells found in many reports might now be attributed to a mitochondrial role in repressing mitochondrial gene expression, which is required for G1 cell cycle progression.” to “**Based on these results we speculate that the decreased proliferative capacity of p107 over expressing cells found in some reports might be attributed to a mitochondrial role. In this case, repressing mitochondrial gene expression, which is required for G1 cell cycle progression.**”

Reviewer #4 (Remarks to the Author):

Bhattacharya et al. report that p107 regulates mitochondrial function and muscle stem cell proliferation fates.

The authors report the following:

1. p107 localizes in the mitochondria of myogenic progenitor cells:

COMMENTS:

Figure 1B: Of the total p107, what’s the percentage that localizes to mitochondria?

As outlined the in updated Supplementary Materials and Methods, using densitometry quantification of total cellular, mitochondrial and cytoplasmic p107 protein by western blot analysis using densitometry on total, mitochondrial and cytoplasmic p107 protein (**new Suppl. Fig. 2**), we found that the percentage in the mitochondria and cytoplasm in asynchronously dividing c2MPs is about 28% and 45% respectively. For these new data we write on page 5 of the Results section the following: “**Of the total cellular p107 protein that was expressed in proliferating c2MPs, about 28% and 45% were present in the mitochondria and cytoplasmic compartments, respectively (Suppl. Fig. 2).**” The Materials and Methods section has been updated for these new findings.

Figure 1C: The image resolution doesn’t allow to finally conclude that p107 and Cox4 colocalize. p107 mitochondrial localization should be performed with MitoTracker Red (which measure mitochondrial mass and membrane potential) and Green (which measures mitochondrial mass regardless of mitochondria activity).

We initially did not use MitoTracker to show co-localization because we had experimental issues with bleeding of the dye into the nucleus. This technical shortcoming has now been surmounted in this revised manuscript and represented by the **new Supplemental Figure 4**. For which we now write on page 6:

“**The protein localization results were corroborated by confocal microscopy and subsequent analysis of generated z-stacks that showed p107 and the mitochondrial protein Cox4 and MitoTracker Red co-localize in c2MPs (Fig. 1E & Suppl. Fig. 4A). The specificity of immunocytochemistry was confirmed by immunofluorescence of p107 in p107 “knockout” c2MPs (p107KO c2MPs) generated by Crispr/Cas9 (Suppl. Fig. 5A). Moreover, p107 and Cox4 or MitoTracker Red fluorescence intensity peaks were matched on a line scanned RGB profile (Fig. 1F & Suppl. Fig. 4B) and orthogonal projection showed co-localization in the XY, XZ, and YZ planes (Fig 1G & Suppl. Fig. 4C). The same assays were used to confirm these findings in proliferating primary wild type (Wt) MPs (prMPs) and p107 genetically deleted prMPs (p107KO prMPs) isolated from Wt and p107KO mouse skeletal muscles, respectively (Fig. 1H,**

1I, 1J & Suppl. Fig. 4D, 4E & 4F). In vivo, in a model of regenerating skeletal muscle caused by cardiotoxin injury, we also found p107 co-localized with Cox4 in proliferating MPs, which express the myogenic stem cell marker Pax7 (**Fig. 1K, 1L & Suppl. Fig. 6**). Together, these data showcase that p107 localizes in the mitochondria of MPs from primary and muscle cell lines, suggesting that it might have an important mitochondrial function in the actively dividing cells.”

For the Mito Tracker experiments we have updated the Materials and methods section.

2. P107 interacts at the mtDNA

COMMENTS:

Figure 2. Control experiments with known mtDNA binding proteins (TFAM, mitochondrial RNA polymerase POLRMT) should be performed.

Unfortunately, we could not perform a positive control ChIP for initiation complex factors Polrmt and Tf2b2m, as mouse specific antibodies capable of immunoprecipitation, which potentially can be used for ChIP are presently not available for these proteins. Nor could we find published reports of have mouse specific antibodies for Polrmt and Tf2b2m used for immunoprecipitation or ChIP been published. Thus, we assessed Tfam interaction at the mtDNA as a potential positive control (**new Fig. 2C**). We find that Tfam is positive in our ChIP assay only during growth arrest when it is mostly present in the mitochondria (**new Fig. 2B & Fig. 2C**) and not during proliferation. For these new data we now write on page 7 of the Results section:

“As p107 does not directly interact with DNA, we assessed potential interacting transcription factors in the mitochondria. We evaluated potential mitochondrial role for the putative mitochondrial transcription factor Tfam, as well as the p107 interacting nuclear transcription factors E2f4 and E2f5³¹. By Western blotting, we found that E2f4 was present in proliferating mitochondrial lysates, compared to negligible levels of Tfam and the complete absence of E2f5 (**Fig. 2B**). Moreover, ChIP analysis indicated that E2f4, but not Tfam, interacted at the mtDNA of proliferating c2MPs, whereas the opposite pattern of interaction was found during growth arrest (**Fig. 2C**). Intriguingly, these results suggest that E2f4 might be a mitochondrial binding partner of p107 at the mtDNA during proliferation of c2MPs.”

The Figure Legends and Materials and Methods sections have been updated with these new findings.

3. NAD⁺/NADH regulates p107 mitochondrial function

COMMENTS:

Figure 3. Beside influencing NAD⁺/NADH ratio, oxamate and DCA affect other metabolites. For instance, DCA increases acetyl-CoA. LDHA knock-down would be more appropriate and would confirm the result obtained with oxamate.

As requested by the reviewer we knocked down LDHa with RNAi (**new Suppl Fig. 12A**). Our new results show that compared to control cells, Ldha knockdown cells had reduced NAD⁺/NADH ratio (**new Suppl Fig. 12B**), increased p107 levels are in the mitochondria (**new Suppl. Fig. 12C**) and as anticipated lower mitochondrial encoded gene expression (**new Suppl. Fig. 12D**). These findings confirm the validity of using oxamate and DCA to increase NADH levels. For these new data we have made slight adjustments to the Results section on page 11:

“We manipulated the cytoplasmic NAD⁺/NADH energy flux in a glucose concentration independent manner to evaluate the importance of the redox potential to p107 function. We

added oxamate (ox) or dichloroacetic acid (DCA) or used RNAi knockdown of *Ldha* (Suppl. Fig. 12A) to decrease the NAD^+/NADH ratio (Fig. 3J, Suppl. Fig. 12B & Suppl. Fig. 13A). Conversely, we also grew cells in the presence of galactose instead of glucose, which increases NAD^+/NADH ratio and cultured cells in varying amounts of glutamine in stripped media, which had no effect on the cytoplasmic NAD^+/NADH ratio (Fig. 3J). As anticipated, Western blot analysis of c2MPs treated with ox, DCA or *Ldha* RNAi showed significantly increased p107 levels in the mitochondria (Fig. 3K, Suppl. Fig. 12C & Suppl. Fig. 13B). On the other hand, galactose treated cells had significantly less p107 in the mitochondria compared to untreated controls and cells treated with glutamine showed no difference in the level of p107 mitochondrial localization (Fig. 3K). These results suggest that p107 mitochondrial localization is based on the cytoplasmic NAD^+/NADH ratio. Mitochondrial gene expression levels with the different treatments corresponded with p107 mitochondrial localization established by the cytoplasmic NAD^+/NADH ratio (Fig. 3L, Suppl. Fig. 12D & Suppl. Fig. 13C). Indeed, significantly higher mitochondrial gene expression levels were observed with the higher cytoplasmic NAD^+/NADH achieved when cells were grown with galactose compared to glucose, whereas significantly lower mitochondrial gene expression was evident when cells were treated with ox or DCA or *Ldha* knockdown that caused a decrease in cytoplasmic NAD^+/NADH (Fig. 3L, Suppl. Fig. 12D & Suppl. Fig. 13C). No significant gene expression changes were present when cells were grown solely with varying amounts of glutamine, except *Nd6*, which is the only mitochondrial gene expressed from the L-strand of mtDNA (Fig. 3L). Together, these results show that p107 acts indirectly as an energy sensor of the cytoplasmic NAD^+/NADH ratio that might influence the potential ATP produced from the mitochondria, because of regulating mitochondrial gene expression.”

The Materials and Methods section has been updated for these new findings.

4. SIRT1 directly regulates p107 mitochondrial function

COMMENTS:

Figure 4A. It is not clear in which cell compartment p107 interacts with SIRT1.

We thank the reviewer for this concern as answering his/her query has strengthened our hypothesis. As requested by the reviewer, we have now performed the Ip/Western for endogenous p107 and Sirt1 in cytoplasmic lysates (new Suppl. Fig. 14), which shows that they interact. This suggests that p107 is regulated by Sirt1 in the cytoplasm. For this new data we write on page 12: “Importantly, reciprocal immunoprecipitation/Western blot analysis for endogenous p107 and Sirt1 in c2MP whole cell as well as cytoplasmic lysates showed that they directly interacted (Fig. 4A & Suppl. Fig. 14). No interactions were apparent in p107KO and Sirt1genetically deleted (Sirt1KO) c2MPs obtained by Crispr/Cas9 (Fig. 4A, Suppl. Fig. 5B & Suppl Fig. 14).” The Materials and Methods section has been updated with these new findings.

Figure 4B. The levels of p107 are greatly increased in Sirt1KO cells. Is this a transcriptional or a protein stabilization effect? Total increase of p107KO in SIRT1KO cells complicates the interpretation of the results related mitochondrial localization of p107.

As per the reviewer’s request we have checked the p107 RNA expression pattern in Sirt1KO cells (new Suppl. Fig. 15A), as well as in nicotinamide treated (new Suppl. Fig. 15C) and in cells treated with 5.5 compared to 25mM glucose cells (new Suppl. Fig. 15B). We found that the

loss of Sirt1 (**Fig. 4B**) or the addition of Sirt inhibitor nicotinamide (**Fig. 4E**) or altering the glucose concentration (that is the NAD⁺/NADH ratio) had no effect on the gene expression of p107. Thus, p107 transcription is not a confounding effect for p107 mitochondrial localization. For these new data we have added to the following sentence “Unlike c2MPs grown in 5.5mM glucose that exhibit relocation of p107 from the mitochondria (**Fig. 3A**), Sirt1KO cells did not exhibit altered p107 mitochondrial localization (**Fig. 4B**)” on page 12 of the Results section: the phrase “nor a change in p107 gene expression, including when grown in different glucose concentrations (**Suppl. Fig. 15A & 15B**).”

We have also altered the sentence on page 13: “We next determined if Sirt1 activity is necessary for p107 mitochondrial function. Inhibition of Sirt1 activity by nicotinamide (nam) increased p107 mitochondrial localization (**Fig. 4E**) that was concomitant with increased mtDNA promoter interaction (**Fig. 4F**).” to read “We next determined if Sirt1 activity is necessary for p107 mitochondrial function. Inhibition of Sirt1 activity by nicotinamide (nam) increased p107 mitochondrial localization (**Fig. 4E**), which was concomitant with increased mtDNA promoter interaction (**Fig. 4F**). This occurred without a change in p107 gene expression (**Suppl. Fig. 15C**).”

Finally, we also assessed p107 gene expression in cells grown in different glucose conditions and found no differences. For this we write on page 9: “The presence of p107 in the mitochondria was inversely associated to the amount in the nucleus (**Fig. 3A**) and was not coupled to a change in p107 gene expression (**Suppl. Fig. 10**)”

For these results we have updated the Materials and Methods section.

Figure 4K. Resveratrol leads to replicative stress and S phase transit and is independent of Sirt1 (Benslimane et al. Molecular Cell DOI:<https://doi.org/10.1016/j.molcel.2020.07.010>) and should not be employed to evaluate Sirt1 function.

As recommended by the reviewer to better ensure specific effects for Sirt1 activation we have appended the resveratrol data to the supplemental data section and replaced it with data by treating cells with known Sirt1 activator srt1720 (**new Figures 4K, 4L, 4M, 4N, 4O and 4P**). We obtained the same results as when an activating concentration (10mM) of resveratrol was used (**new Suppl. Fig. 17**).

For these new data we have removed the following:

“c2MPs grown in a low concentration of res (10mM), had the opposite effect to nam for p107 localization and function. Treatment of c2MPs with this concentration of res decreased p107 within the mitochondria (**Fig. 4K**). The activation of Sirt1 also reduced p107 mtDNA promoter interaction and enhanced the mitochondrial gene expression (**Fig. 4L & 4M**), which corresponded to an increased ATP synthesis rate and capacity of isolated mitochondria (**Fig. 4N & Suppl. Table 1F**). No differences in ATP generation rate and capacity were observed in Sirt1KO c2MPs, which was anticipated with a non-significant consequence on mitochondrial gene expression (**Fig. 4M & 4O**). When Sirt1 activity was repressed by high doses of res, p107 was localized in the mitochondria and gene expression along with ATP generation rate and capacity were significantly decreased (**Suppl. Fig. 7 & Suppl. Table 1G**).”

In its place we have added the following paragraph on page 13: “Next, we activated Sirt1 for a short window of time. Treatment of c2MPs with Sirt1 activator, srt1760, for 3 hours resulted in decreased p107 protein levels within the mitochondria (**Fig. 4K**). The activation of Sirt1 also reduced p107 mtDNA promoter interaction (**Fig. 4L**) and enhanced the mitochondrial

gene expression, but not in p107KO and Sirt1KO cells (**Fig. 4M**). The increase in gene expression with Sirt1 activation within this short time window was associated with increased ATP generation (**Fig. 4N**). Importantly, relative mtDNA copy number was not affected by this short-term treatment (**Fig. 4O**), suggesting that mitochondrial gene expression and not mtDNA copy number contributed to the differences in ATP generation. ATP generation rate and capacity were not altered by srt1760 treatment in Sirt1KO c2MPs (**Fig. 4P**). We also confirmed the importance of Sirt1 activity to p107 mitochondrial function by using resveratrol at low (**Suppl. Fig. 17 & Suppl. Table 1F**) and high doses (**Suppl. Fig. 18 & Suppl. Table 1G**) that indirectly activate and inhibit Sirt1 activity, respectively.”

For these new data we have revised the Figure Legends and Materials and Methods sections.

1. Verona, R., *et al.* E2F activity is regulated by cell cycle-dependent changes in subcellular localization. *Mol Cell Biol* **17**, 7268-7282 (1997).
2. Lindeman, G.J., Gaubatz, S., Livingston, D.M. & Ginsberg, D. The subcellular localization of E2F-4 is cell-cycle dependent. *Proc Natl Acad Sci U S A* **94**, 5095-5100 (1997).
3. Muller, H., *et al.* Induction of S-phase entry by E2F transcription factors depends on their nuclear localization. *Mol Cell Biol* **17**, 5508-5520 (1997).
4. Rodier, G., *et al.* p107 inhibits G1 to S phase progression by down-regulating expression of the F-box protein Skp2. *J Cell Biol* **168**, 55-66 (2005).
5. Zini, N., *et al.* pRb2/p130 and p107 control cell growth by multiple strategies and in association with different compartments within the nucleus. *J Cell Physiol* **189**, 34-44 (2001).
6. Wirt, S.E. & Sage, J. p107 in the public eye: an Rb understudy and more. *Cell Div* **5**, 9 (2010).
7. Hilgendorf, K.I., *et al.* The retinoblastoma protein induces apoptosis directly at the mitochondria. *Genes Dev* **27**, 1003-1015 (2013).
8. Scime, A., *et al.* Rb and p107 regulate preadipocyte differentiation into white versus brown fat through repression of PGC-1alpha. *Cell Metab* **2**, 283-295 (2005).
9. Scime, A., *et al.* Oxidative status of muscle is determined by p107 regulation of PGC-1a. *J Cell Biol* **190**, 651-662 (2010).
10. Porras, D.P., *et al.* p107 Determines a Metabolic Checkpoint Required for Adipocyte Lineage Fates. *Stem Cells* **35**, 1378-1391 (2017).
11. De Sousa, M., Porras, D.P., Perry, C.G., Seale, P. & Scime, A. p107 is a crucial regulator for determining the adipocyte lineage fate choices of stem cells. *Stem Cells* **32**, 1323-1336 (2014).
12. Bhattacharya, D., *et al.* Decreased transcriptional corepressor p107 is associated with exercise-induced mitochondrial biogenesis in human skeletal muscle. *Physiol Rep* **5**(2017).
13. Hsieh, M.C., Das, D., Sambandam, N., Zhang, M.Q. & Nahle, Z. Regulation of the PDK4 isozyme by the Rb-E2F1 complex. *J Biol Chem* **283**, 27410-27417 (2008).
14. Blanchet, E., *et al.* E2F transcription factor-1 regulates oxidative metabolism. *Nature cell biology* **13**, 1146-1152 (2011).

15. Reynolds, M.R., *et al.* Control of glutamine metabolism by the tumor suppressor Rb. *Oncogene* **33**, 556-566 (2014).
16. Nicolay, B.N., *et al.* Proteomic analysis of pRb loss highlights a signature of decreased mitochondrial oxidative phosphorylation. *Genes Dev* **29**, 1875-1889 (2015).
17. Sankaran, V.G., Orkin, S.H. & Walkley, C.R. Rb intrinsically promotes erythropoiesis by coupling cell cycle exit with mitochondrial biogenesis. *Genes Dev* **22**, 463-475 (2008).
18. Ciavarra, G. & Zacksenhaus, E. Rescue of myogenic defects in Rb-deficient cells by inhibition of autophagy or by hypoxia-induced glycolytic shift. *J Cell Biol* **191**, 291-301 (2010).
19. Emanuelsson, O., Brunak, S., von Heijne, G. & Nielsen, H. Locating proteins in the cell using TargetP, SignalP and related tools. *Nat Protoc* **2**, 953-971 (2007).
20. Rahbani, J.F., *et al.* Creatine kinase B controls futile creatine cycling in thermogenic fat. *Nature* **590**, 480-485 (2021).
21. Backes, S., *et al.* Tom70 enhances mitochondrial preprotein import efficiency by binding to internal targeting sequences. *J Cell Biol* **217**, 1369-1382 (2018).
22. Sol, E.M., *et al.* Proteomic investigations of lysine acetylation identify diverse substrates of mitochondrial deacetylase sirt3. *PLoS One* **7**, e50545 (2012).
23. Chang, H.C. & Guarente, L. SIRT1 and other sirtuins in metabolism. *Trends Endocrinol Metab* **25**, 138-145 (2014).
24. Luengo, A., *et al.* Increased demand for NAD(+) relative to ATP drives aerobic glycolysis. *Molecular cell* **81**, 691-707 e696 (2021).
25. McKean, W.B., Toshniwal, A.G. & Rutter, J. A time to build and a time to burn: glucose metabolism for every season. *Molecular cell* **81**, 642-644 (2021).
26. Folmes, C.D., Nelson, T.J., Dzeja, P.P. & Terzic, A. Energy metabolism plasticity enables stemness programs. *Ann N Y Acad Sci* **1254**, 82-89 (2012).
27. Salazar-Roa, M. & Malumbres, M. Fueling the Cell Division Cycle. *Trends Cell Biol* **27**, 69-81 (2017).
28. Mitra, K., Wunder, C., Roysam, B., Lin, G. & Lippincott-Schwartz, J. A hyperfused mitochondrial state achieved at G1-S regulates cyclin E buildup and entry into S phase. *Proc Natl Acad Sci U S A* **106**, 11960-11965 (2009).
29. Wheaton, W.W., *et al.* Metformin inhibits mitochondrial complex I of cancer cells to reduce tumorigenesis. *Elife* **3**, e02242 (2014).
30. Martinez-Reyes, I., *et al.* TCA Cycle and Mitochondrial Membrane Potential Are Necessary for Diverse Biological Functions. *Molecular cell* **61**, 199-209 (2016).
31. Birsoy, K., *et al.* An Essential Role of the Mitochondrial Electron Transport Chain in Cell Proliferation Is to Enable Aspartate Synthesis. *Cell* **162**, 540-551 (2015).
32. Sullivan, L.B., *et al.* Supporting Aspartate Biosynthesis Is an Essential Function of Respiration in Proliferating Cells. *Cell* **162**, 552-563 (2015).
33. Canto, C., Menzies, K.J. & Auwerx, J. NAD(+) Metabolism and the Control of Energy Homeostasis: A Balancing Act between Mitochondria and the Nucleus. *Cell Metab* **22**, 31-53 (2015).
34. Terzioglu, M., *et al.* MTERF1 binds mtDNA to prevent transcriptional interference at the light-strand promoter but is dispensable for rRNA gene transcription regulation. *Cell Metab* **17**, 618-626 (2013).

35. Shutt, T.E., Lodeiro, M.F., Cotney, J., Cameron, C.E. & Shadel, G.S. Core human mitochondrial transcription apparatus is a regulated two-component system in vitro. *Proc Natl Acad Sci U S A* **107**, 12133-12138 (2010).
36. Bouda, E., Stapon, A. & Garcia-Diaz, M. Mechanisms of mammalian mitochondrial transcription. *Protein Sci* **28**, 1594-1605 (2019).
37. Gustafsson, C.M., Falkenberg, M. & Larsson, N.G. Maintenance and Expression of Mammalian Mitochondrial DNA. *Annu Rev Biochem* **85**, 133-160 (2016).
38. Barshad, G., Marom, S., Cohen, T. & Mishmar, D. Mitochondrial DNA Transcription and Its Regulation: An Evolutionary Perspective. *Trends Genet* **34**, 682-692 (2018).
39. She, H., *et al.* Direct regulation of complex I by mitochondrial MEF2D is disrupted in a mouse model of Parkinson disease and in human patients. *J Clin Invest* **121**, 930-940 (2011).
40. Chatterjee, A., *et al.* MOF Acetyl Transferase Regulates Transcription and Respiration in Mitochondria. *Cell* **167**, 722-738 e723 (2016).
41. Park, C.B., *et al.* MTERF3 is a negative regulator of mammalian mtDNA transcription. *Cell* **130**, 273-285 (2007).
42. Wredenberg, A., *et al.* MTERF3 regulates mitochondrial ribosome biogenesis in invertebrates and mammals. *PLoS Genet* **9**, e1003178 (2013).
43. Costa-Machado, L.F. & Fernandez-Marcos, P.J. The sirtuin family in cancer. *Cell Cycle* **18**, 2164-2196 (2019).

REVIEWERS' COMMENTS

Reviewer #1 (Remarks to the Author):

The authors have performed a number of experiments that address the points raised by the Reviewers. I recommend publication.

Reviewer #2 (Remarks to the Author):

The authors have, with great efforts, addressed the issues that have been raised by all 3 reviewers. They have added even more data in a way that this manuscript is now overflowing with data. Although manuscripts can always be improved, I feel it is time to accept this manuscript as it is since all major issues have been addressed.

Reviewer #3 (Remarks to the Author):

The authors have responded adequately to most of my concerns, and given the circumstances I understand their argument regarding the RNA-seq experiment that I asked for. This is the remaining concern:

In their response to concern #1 the authors wrote: "Moreover, ChIP analysis indicated that E2f4, but not Tfam, interacted at the mtDNA of proliferating c2MPs, whereas the opposite pattern of interaction was found during growth arrest (Fig. 2C)." This is a problematic statement. Firstly, the authors briefly state that they have performed ChIP qPCR, focusing on the D-loop (nothing explained about the approach in the Methods section). This experiment is, by definition, limited to the D-loop, suggesting that the authors already assumed that if interaction with the mtDNA indeed occurred, it should have been within the mtDNA promoter region. Firstly, as already stated in the previous round of review, several transcription factors have shown to bind within gene sequences, outside of the D-loop (some are cited by the authors). Secondly, our experience is that D-Loop ChIP signals appear in almost any tested transcription factor, and therefore cannot be easily be interpreted. Third, from some reason the authors do not report any signal for TFAM binding - this is questionable, as it is an HMG protein with little or no preference for certain binding motifs, as it binds across the entire mtDNA (see doi:10.1371/journal.pone.0074513). The authors are asked to tone down their argument about the interacting partners of p107 in the mtDNA. A similar point apply to their response to comment 7, where they state that: "Together these data suggest that p107 has repressor activity when interacting at the mtDNA promoter regulating mtDNA copy number and/or gene expression to reducing the capacity to produce ETC complex subunits, which influences the mitochondria potential for ATP generation".

Response to Reviewer 3

We sincerely thank the Reviewers for recommending our manuscript for publication in Nature Communications. We want to express our gratitude to the Reviewers for taking the time to critically appraise our manuscript. Your input, suggestions and insight has strengthened our findings and made them much more comprehensive.

Below point by point are the answers to the remaining concerns by Reviewer 3.

1) In their response to concern #1 the authors wrote: "Moreover, ChIP analysis indicated that E2f4, but not Tfam, interacted at the mtDNA of proliferating c2MPs, whereas the opposite pattern of interaction was found during growth arrest (Fig. 2C)." This is a problematic statement. Firstly, the authors briefly state that they have performed ChIP qPCR, focusing on the D-loop (nothing explained about the approach in the Methods section). This experiment is, by definition, limited to the D-loop, suggesting that the authors already assumed that if interaction with the mtDNA indeed occurred, it should have been within the mtDNA promoter region.

We agree with the reviewer, our data do not prove that p107 interacts at the D-loop region of mtDNA. We had overlooked this interpretation of the ChIP results. To address his/her previous concerns, we removed all indications that p107 specifically interacted at the D-loop region. We changed the statement on page 7: "We evaluated this potential by performing quantitative chromatin immunoprecipitation (qChIP) analysis on the D-loop regulatory region of isolated mitochondria from c2MPs." to "We evaluated this potential by performing quantitative chromatin immunoprecipitation (qChIP) analysis using primer sets that span the D-loop regulatory region of isolated mitochondria from c2MPs."

2) Third, from some reason the authors do not report any signal for TFAM binding - this is questionable, as it is an HMG protein with little or no preference for certain binding motifs, as it binds across the entire mtDNA (see doi:10.1371/journal.pone.0074513).

Yes, we were also very surprised that we did not detect Tfam interaction at the mtDNA in proliferating c2MPs (c2c12 cells). Corroborating our result, other studies reported very low levels of Tfam interacting at mtDNA in proliferating c2c12 cells (eg. Collu-Marchese, Biosci Rep. 2015 Jun; 35(3): e00221, Barbieri, et al J Aging Res. 2011; 2011: 845379). The amount of Tfam binding to mtDNA is dramatically increased during differentiation (when cells are growth arrested and mitochondrial biogenesis up regulated). We found p107 is expressed in the nucleus early in differentiation and not expressed at all in terminally differentiated myotubes (unpublished data). The lack of Tfam interaction at mtDNA is a very interesting result that we are currently following up.

3) The authors are asked to tone down their argument about the interacting partners of p107 in the mtDNA.

We have toned down our contention about interacting proteins at the mtDNA with p107 by changing the sentence on page 7 from: "Intriguingly, these results suggest that E2f4 might interact with p107 at the mtDNA during proliferation of c2MPs." to "Intriguingly, these results suggest that E2f4 is a possible binding partner for p107 at the mtDNA during proliferation of c2MPs.". Thus, leaving the idea that there might be other possibilities for any potential interacting partners.

4) A similar point apply to their response to comment 7, where they state that: "Together these data suggest that p107 has repressor activity when interacting at the mtDNA promoter regulating mtDNA copy number and/or gene expression to reducing the capacity to produce ETC complex subunits, which influences the mitochondria potential for ATP generation".

We agree, our data do not prove that p107 interacts at the D-loop region of mtDNA. This statement was improperly written. We removed the word "promoter" from the statement on page 9 to read: "Together these data suggest that p107 has repressor activity when interacting at the mtDNA promoter regulating mtDNA copy number and/or gene expression to reducing the capacity to produce ETC complex subunits, which influences the mitochondria potential for ATP generation".

Reviewer #4

The authors have addressed my comments.

The literature refers to what the authors call "actively proliferating MP cell line c2c12 (c2MPs)" as C2C12 myoblasts and I strongly suggest that the latter be employed to avoid any confusion.

We very much thank the reviewer for accepting our answers to his/her concerns. Regarding the additional concern, to avoid any confusion regarding the naming/association of c2MPs with C2C12 myoblasts, we have altered two sentences.

1) On page 5 of the results section we changed the phrase "...p107 was found in the cytoplasm of actively proliferating MP cell line c2c12 (c2MPs), as is the case in other cell types³¹ (Fig. 1a)." to read "*... p107 was found in the cytoplasm of C2C12 myoblasts, an actively proliferating MP cell line we designated as "c2MPs" (Fig. 1a).*

2) On page 23 of the Methods section, we changed the phrase "The c2c12 myogenic progenitor cell (c2MP) line was purchased from the American Tissue Type Culture (ATTC) ..." to "*The C2C12 myoblast cell line (designated as c2MPs) was purchased from the American Tissue Type Culture (ATTC)....*"